# Nuclear receptor Ftz-f1 promotes follicle maturation and ovulation partly via bHLH/PAS transcription factor Sim

Elizabeth M Knapp[1], Wei Li[1], Vijender Singh[2], Jianjun Sun[1,2]*

[1]Department of Physiology & Neurobiology, University of Connecticut, Storrs, United States; [2]Institute for Systems Genomics, University of Connecticut, Storrs, United States

**Abstract** The NR5A-family nuclear receptors are highly conserved and function within the somatic follicle cells of the ovary to regulate folliculogenesis and ovulation in mammals; however, their roles in *Drosophila* ovaries are largely unknown. Here, we discover that Ftz-f1, one of the NR5A nuclear receptors in *Drosophila*, is transiently induced in follicle cells in late stages of oogenesis via ecdysteroid signaling. Genetic disruption of Ftz-f1 expression prevents follicle cell differentiation into the final maturation stage, which leads to anovulation. In addition, we demonstrate that the bHLH/PAS transcription factor Single-minded (Sim) acts as a direct target of Ftz-f1 to promote follicle cell differentiation/maturation and that Ftz-f1's role in regulating Sim expression and follicle cell differentiation can be replaced by its mouse homolog steroidogenic factor 1 (mSF-1). Our work provides new insight into the regulation of follicle maturation in *Drosophila* and the conserved role of NR5A nuclear receptors in regulating folliculogenesis and ovulation.

*For correspondence:
jianjun.sun@uconn.edu

**Competing interests:** The authors declare that no competing interests exist.

## Introduction

Female fertility, an essential half of the reproductive equation, requires proper follicle maturation and ovulation. The NR5A family of nuclear receptors are critical for the success of these complex ovarian processes across species (*Jeyasuria et al., 2004*; *Meinsohn et al., 2019*; *Mlynarczuk et al., 2013*; *Sun and Spradling, 2013*; *Suresh and Medhamurthy, 2012*). The majority of what is known concerning these NR5A receptors in female fertility stems from studies performed over the past two decades in rodent models. These investigations have shown that both members of this family, NR5A1 (steroidogenic factor-1 or SF-1) and NR5A2 (liver receptor homolog-1 or LRH-1), are expressed in the follicle cells that encapsulate the oocyte throughout oogenesis (*Falender et al., 2003*; *Hinshelwood et al., 2003*). Follicle-cell-specific loss of either receptor leads to drastically impaired fertility. LRH-1 knockout in granulosa cells in either primary or more developed antral follicles results in severe anovulation, which is attributed to the inhibition of —cumulus expansion, expression of steroidal biosynthetic genes, and granulosa cell proliferation/differentiation (*Bertolin et al., 2014*; *Bertolin et al., 2017*; *Bianco et al., 2019*; *Duggavathi et al., 2008*; *Meinsohn et al., 2018*). Targeted depletion of SF-1 in granulosa cells of primary follicles has shown to result in hypoplastic ovaries and a dramatically reduced number of developing follicles (*Pelusi et al., 2008*). Much less is known about the molecular mechanism of SF-1 in these ovarian follicle cells.

SF-1 was initially recognized as the mammalian homolog of the *Drosophila fushi tarazu-factor 1* (*ftz-f1*), which was first identified as a transcription factor binding to the promoter of the pair-rule segmentation gene *fushi tarazu* (*ftz*) during early embryogenesis (*Lala et al., 1992*; *Ueda et al., 1990*). *Drosophila ftz-f1* gene encodes two protein isoforms (αFtz-f1 and βFtz-f1), each comprised of

**eLife digest** When animals reproduce, females release eggs from their ovaries which then get fertilized by sperm from males. Each egg needs to properly mature within a collection of cells known as follicle cells before it can be let go. As the egg matures, so do the follicle cells surrounding it, until both are primed and ready to discharge the egg from the ovary. Mammals rely on a protein called SF-1 to mature their follicle cells, but it is unclear how this process works.

Most animals – from humans to fruit flies – release their eggs in a very similar way, using many of the same proteins and genes. For example, the gene for SF-1 in mammals is similar to a gene in fruit flies which codes for another protein called Ftz-f1. Since it is more straightforward to study ovaries in fruit flies than in humans and other mammals, investigating this protein could shed light on how follicle cells mature. However, it remained unclear whether Ftz-f1 plays a similar role to its mammalian counterpart.

Here, Knapp et al. show that Ftz-f1 is present in the follicle cells of fruit flies and is required for them to properly mature. Ftz-f1 controlled this process by regulating the activity of another protein called Sim. Further experiments found that the gene that codes for the SF-1 protein in mice was able to compensate for the loss of Ftz-f1 and drive follicle cells to mature.

Studying how ovaries release eggs is an essential part of understanding female fertility. This work highlights the similarities between these processes in mammals and fruit flies and may help us understand how ovaries work in humans and other mammals. In the future, the findings of Knapp et al. may lead to new therapies for infertility in females and other disorders that affect ovaries.

unique N-terminal sequences and common C-terminal sequences (*Lavorgna et al., 1991*; *Lavorgna et al., 1993*). αFtz-f1 is maternally supplied and functions as a cofactor for Ftz during early embryogenesis (*Guichet et al., 1997*; *Yu et al., 1997*). On the other hand, βFtz-f1 is only transiently induced after each ecdysone pulse in the late embryo, larvae, and pupae, and functions as a competency factor for stage-specific responses to ecdysone pulses and progression into the next developmental stages (*Broadus et al., 1999*; *Cho et al., 2014*; *Lavorgna et al., 1993*; *Woodard et al., 1994*). In addition, βFtz-f1 precisely controls the timing of ecdysone pulses through regulating ecdysteroid synthesis enzymes (*Akagi et al., 2016*; *Parvy et al., 2005*; *Talamillo et al., 2013*). Therefore, βFtz-f1 is essential for late embryogenesis, larval molting, metamorphosis, and pupal development (*Bond et al., 2011*; *Boulanger et al., 2011*; *Sultan et al., 2014*; *Yamada et al., 2000*). Ftz-f1 has also been found to function as an oncogene and promote tumorigenesis and tumor invasiveness in *Drosophila* imaginal discs (*Atkins et al., 2016*; *Külshammer et al., 2015*; *Song et al., 2019*). Even though initial studies demonstrated the potential for Ftz-f1 in adult tissues (*Ueda et al., 1990*), little has been done to study what roles Ftz-f1 plays in adult flies, particularly in oogenesis.

*Drosophila* oogenesis is an excellent model for studying many cell biology questions in the last few decades. *Drosophila* oogenesis occurs in the ovariole, ~16 of which bundle together to form an ovary. At the anterior tip of the ovariole, germline and follicle stem cells proliferate to produce daughter cells to form a stage-1 egg chamber (also named follicle in this paper), which develop through 14 morphologically distinct stages into a stage-14 egg chamber [also named mature follicle; (*Spradling, 1993*). Each follicle contains a layer of somatic follicle cells encasing 16 interconnected germ cells, one of which differentiates into an oocyte, while the rest become nurse cells to support oocyte growth and are eventually degraded in mature follicles. Somatic follicle cells proliferate at stages 1–6 and transition into endoreplication at stages 7-10A induced by Notch signaling (*Klusza and Deng, 2011*). At stage 10B, a pulse of ecdysone signaling induces follicle cell transition from endoreplication to synchronized gene amplification via zinc-finger transcription factor Ttk69 (*Sun et al., 2008*). This is also accompanied by the downregulation of the zinc-finger transcription factor Hindsight (Hnt) and the upregulation of the homeodomain transcription factor Cut in stage-10B follicle cells. As follicles develop from stage 10B onwards, Ttk69 and Cut are diminished. By stage 14, another critical follicle cell transition occurs, accompanied by re-upregulation of Hnt and complete loss of Cut and Ttk69 (*Knapp et al., 2019*). This transition is critical for the follicle to gain ovulatory competency via upregulation of Octopamine receptor in mushroom body (Oamb) and Matrix metalloproteinase 2 (Mmp2) (*Deady and Sun, 2015*; *Deady et al., 2015*; *Deady et al., 2017*;

*Knapp et al., 2019*). In addition, stage-14 follicle cells upregulate NADPH oxidase (Nox) expression, downregulate EcR.B1 and EcR.A, and receive another ecdysteroid signaling via EcR.B2 to become ovulatory competent (*Knapp and Sun, 2017*; *Li et al., 2018*). However, it is largely unknown how follicle cells differentiate from stage 10B to stage 14.

In this study, we demonstrate that Ftz-f1 is transiently expressed in *Drosophila* follicle cells at stages 10B-12 and this expression is induced by ecdysteroid signaling in stage-10B follicle cells, independent of Ttk69. Loss of *ftz-f1* in follicle cells after stage 10B severely inhibits follicle cell differentiation into the final maturation stage, resulting in follicles incompetent for follicle rupture and ovulation. In addition, we identify the basic helix-loop-helix/PAS (bHLH/PAS) transcription factor Single-minded (Sim), whose functions are known in the central nervous system development (*Crews et al., 1988*; *Muralidhar et al., 1993*; *Nambu et al., 1990*; *Thomas et al., 1988*), functioning downstream of Ftz-f1 for follicle cell differentiation/maturation. RNA-seq and CUT&RUN analyses (*Meers et al., 2019*; *Zhu et al., 2019*; *Skene and Henikoff, 2017*) suggest that Sim is a direct target of Ftz-f1 in follicle cells. Furthermore, we demonstrate the role of Ftz-f1 in follicle cell maturation is functionally conserved as ectopic expression of mouse SF-1 is able to rescue Ftz-f1's function in this process. These findings demonstrate a more conserved role of NR5A nuclear receptors in *Drosophila* and mammalian reproduction and help elucidate potential mechanisms downstream of NR5A nuclear receptor signaling required for female fertility across species.

## Results

### Ftz-f1 expression is induced in stage-10B follicle cells through ecdysteroid signaling

To investigate the role of Ftz-f1 in female fertility, we first analyzed the expression of Ftz-f1 throughout oogenesis using anti–Ftz-f1 antibody. Ftz-f1 protein is not detected in germline cells and ovarian follicle cells from stage 1 to stage 10A (*Figure 1A*); however, it is drastically upregulated in all follicle cells at stage 10B (*Figure 1B*), when follicle cells transition into synchronized gene amplification. Following stage 10B, Ftz-f1 begins to progressively decrease in follicle cells (except anterior stretch follicle cells) and is no longer detectable in stage-13/14 follicle cells (*Figure 1C–F*). A *ftz-f1::GFP.FLAG* transgene showed that the expression of Ftz-f1::GFP.FLAG tagged protein completely matches Ftz-f1 antibody staining (*Figure 1—figure supplement 1A–E*). In addition, we also examined the *ftz-f1* transcription using the enhancer trap line *ftz-f1* *fs(3)2877*, which has a P-element containing *lacZ* gene inserted in the *ftz-f1* gene (*Karpen and Spradling, 1992*). Expression of βGal is also induced in stage-10B follicle cells and stays high in stage-13/14 follicle cells (*Figure 1—figure supplement 1F–J*), which is likely a result of βGal not being subjected to endogenous protein regulation. Together, our data suggest that both *ftz-f1* mRNA and protein are transiently induced in stage-10B to 12 follicle cells during *Drosophila* oogenesis.

The drastic upregulation of Ftz-f1 at stage 10B is concurrent to the ecdysone-induced transition from endoreplication to gene amplification at stages 10A/10B, which is mediated by the upregulation of the zinc-finger transcription factor Ttk69 (*Sun et al., 2008*). Therefore, we examined whether ecdysone signaling induces *ftz-f1* expression in follicle cells. Using the flip-out Gal4 system (*Pignoni and Zipursky, 1997*), we disrupted the ecdysone signaling via misexpressing a dominant-negative (DN) form of ecdysone receptor (EcR$^{DN}$) (*Cherbas et al., 2003*). EcR$^{DN}$-overexpressing follicle cells showed a complete loss of Ftz-f1 in stage-10B egg chambers (*Figure 1G*), indicating that Ftz-f1 expression is induced by ecdysone signaling. We also investigated whether premature activation of ecdysone signaling in follicle cells was sufficient to induce premature Ftz-f1 expression. Treating egg chambers with exogenous 20-hydroxyecdysone (20E) is able to prematurely activate the EcR ligand sensor in follicle cells prior to stage 10 (*Sun et al., 2008*; *Figure 1—figure supplement 2A*) but is not sufficient to induce premature expression of Ftz-f1 (*Figure 1—figure supplement 2B*). Previous work also showed that Ftz-f1 is only induced during low ecdysone titer. Manipulation of *Cyp18a1,* encoding a cytochrome P450 enzyme involved in lowering 20E titer, influences Ftz-f1 expression during the prepupa-to-pupa transition (*Rewitz et al., 2010*). In contrast, neither ectopic expression nor knockdown of *Cyp18a1* in follicle cells was able to affect Ftz-f1 expression (*Figure 1—figure supplement 2C–F*). Altogether, our data suggest that Ftz-f1 expression in stage-10B follicle cells is induced by ecdysone signaling and seems insensitive to the ecdysone level.

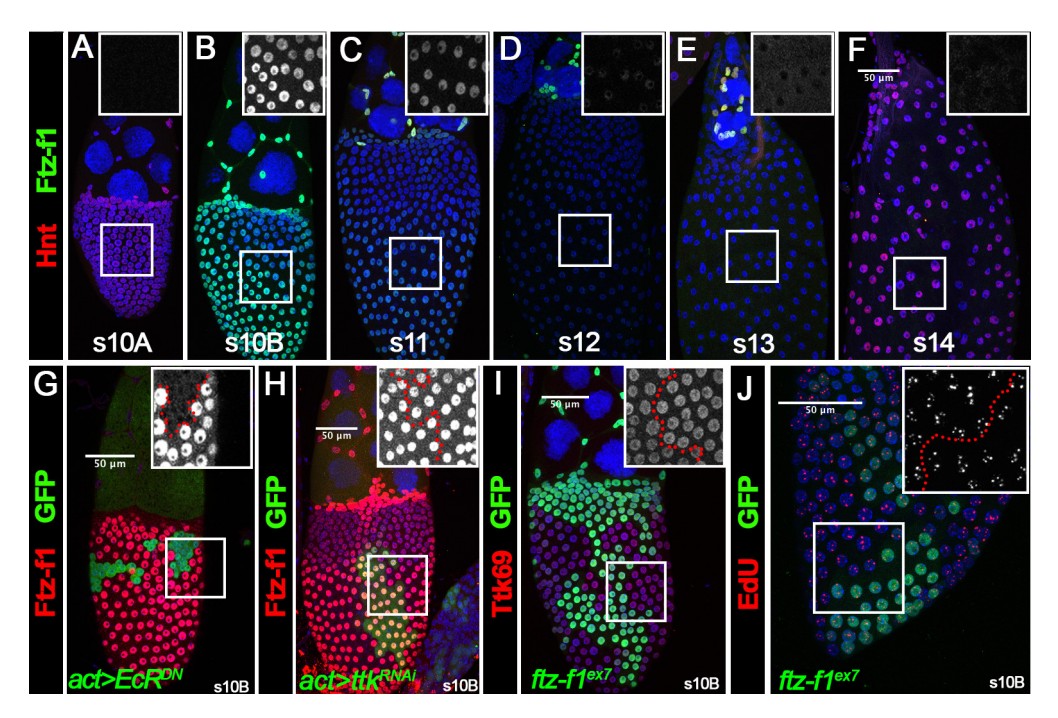

**Figure 1.** Ftz-f1 is induced in stage-10B follicle cells through ecdysteroid signaling. (**A–F**) The expression of Ftz-f1 protein in late oogenesis. Ftz-f1 protein is detected by anti–Ftz-f1 antibody shown in green. Hnt expression (shown in red) is used to mark stage-10A (**A**) and stage-14 (**F**) follicles. The insets are higher magnification of Ftz-f1 expression (white) in outlined areas. All images from A-F are acquired using the same microscopic settings. (**G–H**) Ftz-f1 protein expression (red in G-H) in stage-10B egg chambers with flip-out Gal4 clones (marked by green GFP in G-H) overexpressing $EcR^{DN}$ (act >$EcR^{DN}$ in G) or $ttk^{RNAi}$ (act >$ttk^{RNAi}$ in H). Insets show higher magnification of Ftz-f1 expression in squared area. The clone boundary is outlined by red dashed line. (**I–J**) Ttk69 expression (red in I) and EdU staining (red in J) in stage-10B egg chambers with $ftz$-$f1^{ex7}$ mutant follicle cell clones (marked by green GFP). Insets show the higher magnification of Ttk69 expression (I) and EdU staining (J) in squared areas with the clone boundary marked by red dashed line. Nuclei are marked by DAPI in blue in all figures.

The online version of this article includes the following figure supplement(s) for figure 1:

**Figure supplement 1.** Expression pattern of $ftz$-$f1$::GFP.FLAG and $ftz$-$f1$-lacZ in late oogenesis.

**Figure supplement 2.** Ftz-f1 expression is not sensitive to the ecdysone level.

**Figure supplement 3.** Ttk69 and Sim are efficiently knocked down by overexpression of $ttk^{RNAi}$ and $sim^{RNAi}$, respectively.

To determine whether Ftz-f1 is induced by Ttk69, the downstream target of ecdysone signaling, we knocked down Ttk69 expression by overexpressing $ttk^{RNAi}$ in the flip-out Gal4 clones. Follicle-cell clones with $ttk^{RNAi}$ overexpression showed no detectable Ttk69 (*Figure 1—figure supplement 3A*) but normal Ftz-f1 expression in stage-10B egg chambers (*Figure 1H*). To determine whether Ftz-f1 regulates Ttk69 expression, we generated $ftz$-$f1^{ex7}$ mutant clones using the MARCM system (*Wu and Luo, 2006*). $ftz$-$f1$ mutant follicle cells exhibited normal expression of Ttk69 (*Figure 1I*). In addition, $ftz$-$f1$ mutant follicle cells properly transitioned into the gene amplification stage according to punctate EDU staining (*Figure 1J*). Our results indicate that ecdysone signaling induces both Ftz-f1 and Ttk69 upregulation in stage-10B follicle cells; the latter one leads to the endoreplication/gene amplification transition, while the former one does not.

## Transient expression of Ftz-f1 in late oogenesis is required for ovulation and follicle rupture

To determine the function of Ftz-f1 in follicle cells, we knocked down $ftz$-$f1$ expression in follicle cells using *Vm26Aa-Gal4*, which starts to express in all follicle cells (except anterior stretch follicle cells) at stage 10 (*Peters et al., 2013*). Both $ftz$-$f1^{RNAi1}$ and $ftz$-$f1^{RNAi2}$ showed efficient knockdown of $ftz$-$f1$

in stage-10B and stage-12 follicle cells when driven by *Vm26Aa-Gal4*, although *ftz-f1RNAi1* is more efficient than *ftz-f1RNAi2* (*Figure 2A–C*, *Figure 2—figure supplement 1A–C*). Females with such genetic manipulation (named *ftz-f1RNAi* females) laid significantly fewer eggs than control females (*Figure 2D* and *Figure 2—figure supplement 1D*). In addition, *ftz-f1RNAi1* females showed a severe

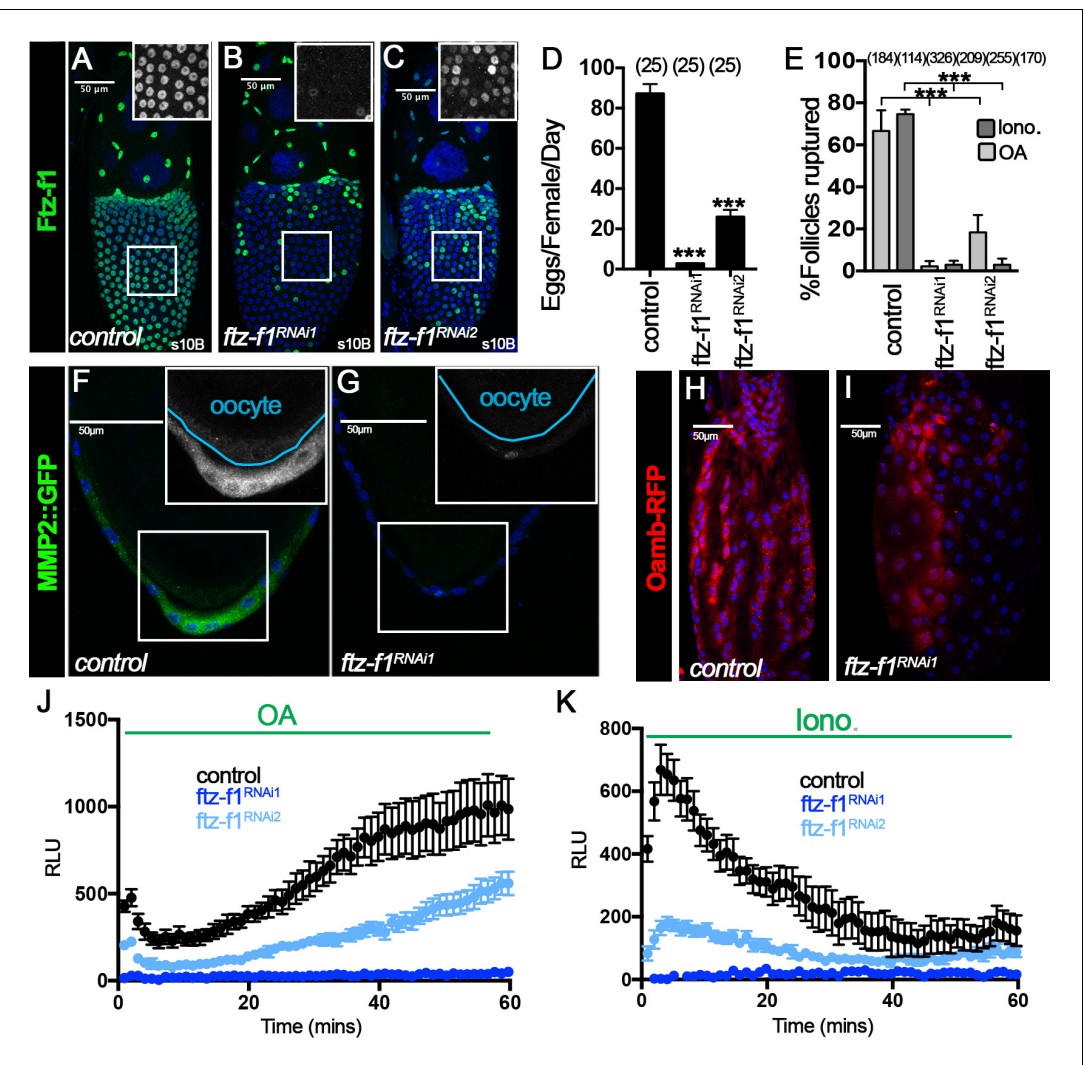

**Figure 2.** Ftz-f1 is required for ovulation and follicle rupture. (**A–C**) Representative images show Ftz-f1 protein expression (green in A-C) in stage-10B egg chambers of control (**A**), *ftz-f1RNAi1* (**B**), and *ftz-f1RNAi2* (**C**) females with *Vm26Aa-Gal4*. The insets are higher magnification of Ftz-f1 expression in squared areas. (**D**) Quantification of egg laying in control or *ftz-f1RNAi* females with *Vm26Aa-Gal4* and *Oamb-RFP*. The number of females is noted above each bar. (**E**) Quantification of OA-induced (light grey bars) and Ionomycin-induced (dark grey bars) follicle rupture using mature follicles isolated from control or *ftz-f1RNAi* females with *Vm26Aa-Gal4* and *Oamb-RFP*. Mature follicles were isolated according to *Oamb-RFP* expression. The number of mature follicles analyzed is noted above each bar. (**F–G**) Representative images show Mmp2::GFP expression (green in F-G) in stage-14 egg chambers from control (**F**) or *ftz-f1RNAi 1* (**G**) females with *Vm26Aa-Gal4*. Insets show higher magnification of Mmp2::GFP expression in posterior follicle cells in squared areas. Oocytes are outlined in cyan. (**H–I**) Representative images show *Oamb-RFP* expression (red) in stage-14 egg chambers from control (**H**) and *ftz-f1RNAi 1* (**I**) females with *Vm26Aa-Gal4*. (**J–K**) Quantification L-012 luminescent signal (indicating superoxide production) in stage-14 egg chambers from control (black), *ftz-f1RNAi1* (dark blue), and *ftz-f1RNAi2* (light blue) females with *VM26Aa-Gal4* and *Oamb-RFP*. Follicles are either stimulated with OA (**J**) or Ionomycin (**K**). Nuclei are marked by DAPI in blue. ***p<0.001 (Student's t-test).

The online version of this article includes the following figure supplement(s) for figure 2:

**Figure supplement 1.** *ftz-f1* knockdown causes defects in ovulation and egg morphology.

retention of stage-14 follicles inside their ovaries (*Figure 2—figure supplement 1E*), indicating an ovulation defect.

To support this observation, we examined whether stage-14 follicles from *ftz-f1^RNAi* females are competent to Octopamine (OA)-induced follicle rupture (*Deady and Sun, 2015*; *Knapp et al., 2018*). Using the *47A04-LexA* driving *LexAop2-6XGFP* as a reporter for isolating mature follicles, we found that mature follicles from control females had ~83% of follicles ruptured after OA stimulation, consistent with our previous result (*Deady and Sun, 2015*). In contrast, mature follicles from *ftz-f1^RNAi1* and *ftz-f1^RNAi2* females showed 6% and 17% follicle rupture, respectively (*Figure 2—figure supplement 1F*). Since hexameric GFP showed punctate GFP signal in mature follicle cells (*Figure 2—figure supplement 1J–L*), we also used *Oamb-RFP* as a reporter for isolating mature follicles from both control and *ftz-f1^RNAi* females to perform OA-induced follicle rupture. We observed 67% follicle rupture from control females, but 2% and 18% follicle rupture from *ftz-f1^RNAi1* and *ftz-f1^RNAi2* females, respectively (*Figure 2E* and *Figure 2—figure supplement 1G–I*). All the data suggest that expression of Ftz-f1 in follicle cells from stage 10B to stage 12 is required for follicle rupture and ovulation.

Our recent work has demonstrated that OA/Oamb signaling leads to calcium influx, which activates both Mmp2 and Nox to regulate follicle rupture (*Deady and Sun, 2015*; *Li et al., 2018*). To determine what is defective in follicles from *ftz-f1^RNAi* females, we first examined whether ionomycin, a $Ca^{2+}$ ionophore, is sufficient to induce these follicles to rupture. Mature follicles from control females showed 75% follicle rupture with ionomycin stimulation; however, mature follicles from *ftz-f1^RNAi* females only showed ~3% follicle rupture (*Figure 2E*). Similar results were also found when mature follicles were isolated according to *LexAop2-6XGFP* (*Figure 2—figure supplement 1F*). The incompetency of ionomycin to induce follicle rupture in follicles isolated from *ftz-f1^RNAi* females suggests that either components downstream of the calcium rise or ovulatory genes parallel to the calcium pathway are defective in these follicles. Consistent with this, we found that Mmp2 expression in posterior follicle cells was completely disrupted in stage-14 follicles from *ftz-f1^RNAi* females (*Figure 2F–G*). In addition, we found that these follicles were defective in OA-induced and ionomycin-induced superoxide production (*Figure 2J–K*), indicating that Nox expression might also be disrupted in mature follicles of *ftz-f1^RNAi* females. Furthermore, we noticed that *Oamb-RFP* expression became patchy in mature follicles of *ftz-f1^RNAi* females when examined in higher magnification, indicating that *Oamb* expression is also disrupted (*Figure 2H–I*). Follicles from *ftz-f1^RNAi* females also exhibited morphological defects in overall shape and dorsal appendage formation (*Figure 2—figure supplement 1M–O*). Altogether, these results indicate that expression of Ftz-f1 in stage-10B–12 follicle cells is essential for follicles to mature and become competent to OA-induced follicle rupture and ovulation.

## Ftz-f1 promotes follicle cell differentiation into the final maturation stage

We have recently demonstrated that follicle cells experience a novel transition from stage 13 to 14 by downregulation of Cut and Ttk69 and upregulation of Hnt, which promotes Oamb and Mmp2 expression and follicle maturation (*Deady et al., 2017*; *Knapp et al., 2019*). Analysis of Hnt expression in stage-14 follicles from *ftz-f1^RNAi* females revealed a patchy expression of Hnt that overlaps with *Oamb-RFP* expression (*Figure 3—figure supplement 1A–B*). In addition, Cut and Ttk69 were still detected in follicle cells without *Oamb-RFP* (*Figure 3—figure supplement 1C–F*), consistent with the fact that Cut antagonizes Hnt expression in stage-14 follicle cells (*Knapp et al., 2019*). The patchy nature of follicle cell markers is likely due to the incomplete knockdown of *ftz-f1* using RNAi. All these data support the hypothesis that *ftz-f1* is required for follicle cells to transition into the final maturation stage.

To determine whether Ftz-f1 functions cell-autonomously in follicle cell differentiation, we generated *ftz-f1* mutant follicle-cell clones. Consistent with our hypothesis, *ftz-f1* mutant clones did not upregulate Hnt expression and continued to express Cut and Ttk69 in stage-14 follicles in a cell-autonomous fashion (*Figure 3A–C*). In addition, EcR.A and EcR.B1, two isoforms downregulated in wild-type stage-14 follicle cells, were still detected at the high level in *ftz-f1* mutant follicle cells (*Figure 3D–E*). Furthermore, we also found that another zinc-finger transcription factor Broad-Complex (Br-C; *DiBello et al., 1991*) was downregulated in wild-type follicle cells but remained high in *ftz-f1* mutant follicle cells (*Figure 3F*). Finally, *ftz-f1* mutant follicle cells continue to have punctate

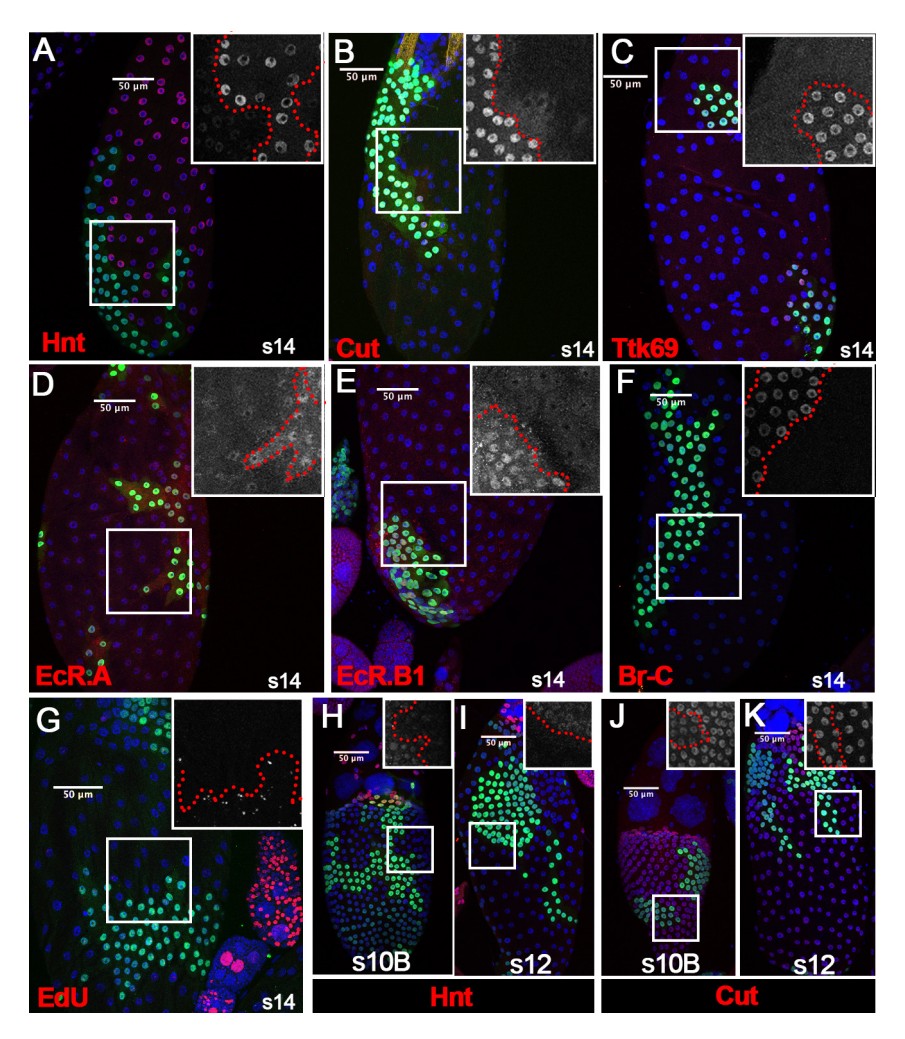

**Figure 3.** Ftz-f1 promotes follicle cell differentiation into the final maturation stage. (A–F) Representative images show the expression of Hnt (A), Cut (B), Ttk69 (C), EcR.A (D), EcR.B1 (E), and Br-C (F) in stage-14 egg chambers with *ftz-f1^ex7* mutant follicle cell clones (marked by green GFP). Insets show higher magnification of Hnt (A), Cut (B), Ttk69 (C), EcR.A (D), EcR.B1 (E), and Br-C (F) in squared areas with the clone boundary marked by red dashed line. (G) Edu staining (red in G) in stage-14 egg chambers with *ftz-f1^ex7* mutant follicle cell clones (marked by green GFP). The inset shows the higher magnification of Edu staining. (H–I) Hnt expression (red in H-I) in stage-10B (H) and stage-12 (I) egg chambers with *ftz-f1^ex7* mutant follicle cell clones (marked by green GFP). Insets show the higher magnification of Hnt expression (H–I). (J–K) Cut expression (red in J-K) in stage-10B (J) and stage-12 (K) egg chambers with *ftz-f1^ex7* mutant follicle cell clones (marked by green GFP). Insets show the higher magnification of Cut expression (J–K) in squared areas with the clone boundary marked by red dashed line. Nuclei are marked by DAPI in blue.

The online version of this article includes the following figure supplement(s) for figure 3:

**Figure supplement 1.** ftz-f1 knockdown disrupts follicle cell transition into stage 14.
**Figure supplement 2.** Analysis of Cut expression in *ftz-f1* mutant clones.

EDU staining, while neighboring wild-type follicle cells have already ceased gene amplification in stage 14 (*Figure 3G*).

To determine which stages *ftz-f1* mutant follicle cells were arrested in, we carefully examined Hnt and Cut expression in *ftz-f1* mutant clones from stage 10B to stage 13. Previous work showed that Hnt is undetectable at the end of stage 10B, while Cut is fully upregulated (*Sun et al., 2008*). Indeed, we found that Hnt was downregulated in *ftz-f1* mutant clones at stage 10B; however, Hnt expression was not fully diminished in *ftz-f1* mutant clones at stage 10B or stage 12 (*Figure 3H–I*). In

addition, Cut expression was upregulated in *ftz-f1* mutant clones at stage 10B, but it was not upregulated as high as that in neighboring wild-type follicle cells (*Figure 3J* and *Figure 3—figure supplement 2A*). This difference was undetectable at stage 12 when Cut is reduced in wild-type follicle cells (*Figure 3K* and *Figure 3—figure supplement 2B–D*). Altogether, these data suggest that *ftz-f1* mutant follicle cells were arrested at the end of stage 10B. Therefore, ecdysone-induced Ftz-f1 functions cell-autonomously to promote follicle cell differentiation and progression into the final stages of maturation.

## bHLH/PAS transcription factor Sim is a direct target of Ftz-f1 in stage-10B follicle cells

To understand how Ftz-f1 promotes follicle cell differentiation in late oogenesis, we tried to identify the direct targets of Ftz-f1. We first performed RNA-seq analysis using hand-dissected stage-10B–12 follicles from control and *ftz-f1^RNAi1^* females with *Vm26Aa-Gal4*. Principle component analysis clearly showed separation of control samples from *ftz-f1^RNAi1^* samples (*Figure 4A*). DE-seq analysis identified 197 downregulated genes and 192 upregulated genes that had more than two-fold change in expression level and adjusted p-value less than 0.01 (*Figure 4B* and *Supplementary file 1*). It is worth noting that neither *hnt* nor *cut* and *ttk* are among the differentially expressed genes.

To profile the Ftz-f1-binding sites throughout the genome in follicle cells, we carried out CUT&RUN experiment, an assay utilizing transcription factor-specific antibody to bring micrococcal nuclease (MNase) to release transcription factor-bound short fragments in intact cells followed by next-generation sequencing (*Meers et al., 2019*; *Skene and Henikoff, 2017*). We implemented the CUT&RUNTools workflow developed by Yuan's group with minor modification (see materials and methods; *Zhu et al., 2019*). With highly stringent criteria, we identified 520, 943, and 550 narrow peaks in three biological replicates, respectively. All three biological replicates showed similar peak distribution throughout the genome (*Figure 4—figure supplement 1A–C*). Majority of the peaks are located within 3 kb of transcriptional start site (TSS; *Figure 4—figure supplement 1C*), consistent with the idea that Ftz-f1 is a transcriptional regulator. Using MEME-chip (*Machanick and Bailey, 2011*), de novo motif search with sequences flanking the peak summit identified similar motifs (CAAGGTCARV for replicate 1, CAAGGTCR for replicate 2, and DBTCAAGGTCA for replicate 3; *Figure 4C*), which are also similar to the canonical Ftz-f1 binding motif YCAAGGYCR (*Murata et al., 1996*; *Ueda et al., 1990*). Footprinting analysis for all three motifs showed the typical pattern of a high posterior probability of cut (or cut-frequency) in the motif flanking region and a low posterior probability of cut in the motif core (*Figure 4D*), presumably due to the protection of transcription factor-bound DNA. In addition, all three motifs showed a symmetric motif footprint profile (*Figure 4D*). Altogether, these data suggest that de novo-identified motifs are the true Ftz-f1-binding motifs. In total, we identified 166, 505, and 389 motif sites within the narrow peaks in each replicate, respectively (*Supplementary file 2*). The nearest gene/transcript associated with each motif site were also identified using ChIPseeker (*Yu et al., 2015*) and listed in *Supplementary file 2*.

To identify the direct target genes of Ftz-f1 in follicle cells, we set the following criteria: (1) the gene must be differentially expressed according to the RNA-seq analysis; and (2) the gene must contain a direct Ftz-f1 binding site, which is defined as a site containing overlapping Ftz-f1-binding motifs appeared in at least two of the three biological replicates and with a binding log-odds score >5. The log-odds score is a binding probability score that quantifies the similarity between the cuts at each motif occurrence and the aggregate footprint pattern (*Zhu et al., 2019*). With these criteria, we identified 15 genes/transcripts that were likely direct targets of Ftz-f1 (*Supplementary file 3*). Among these genes, 13 were downregulated genes and 2 were upregulated genes. Only two of the genes (*Eip74EF* and *sim*) encode transcription factors. *Eip74EF* (*Ecdysone-induced protein 74EF*) encodes a transcription factor that responds to different concentration of 20E during puparium formation (*Burtis et al., 1990*), while *sim* (*single-minded*) encodes a bHLH/PAS-domain transcription factor in embryonic neuronal development (*Crews et al., 1988*; *Nambu et al., 1990*; *Thomas et al., 1988*).

To understand how Ftz-f1 promotes follicle cell differentiation in late oogenesis, we focused on the bHLH/PAS transcription factor Sim for the following reasons: 1) transcription factors will make profound changes during cell differentiation; 2) *sim* was identified in an ongoing genetic screen for *Drosophila* ovulatory genes; and 3) only one single peak containing Ftz-f1-binding site was clearly

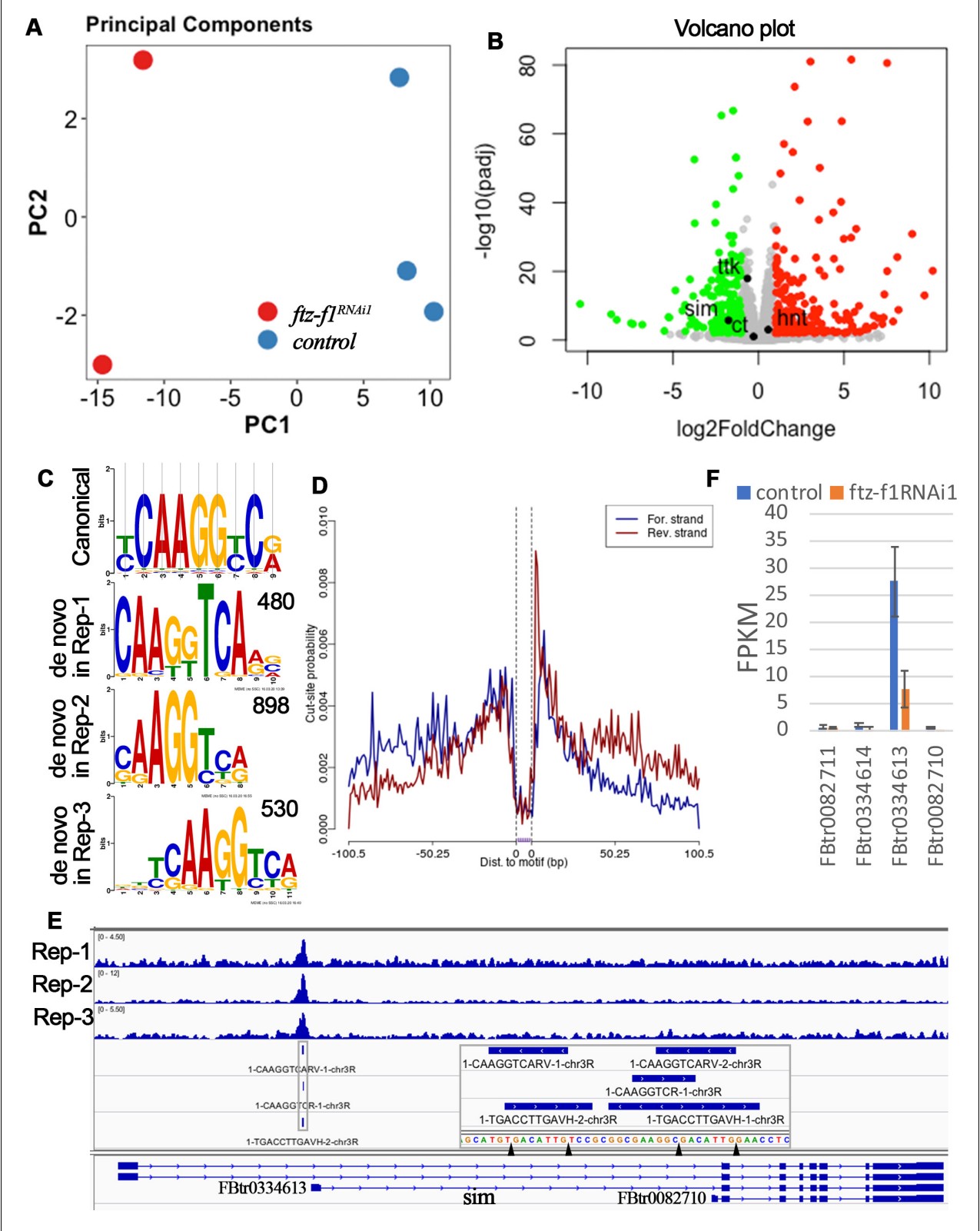

**Figure 4.** RNA-seq and CUT&RUN analyses indicate *sim* as a direct target of Ftz-f1. (**A**) Principle component analysis of RNA-seq data. (**B**) A volcano plot shows the differentially expressed genes between control and *ftz-f1^RNAi^* egg chambers. The significantly upregulated and downregulated genes were marked red and green, respectively. (**C**) The comparison of de novo-identified Ftz-f1-binding motifs and the canonical Ftz-f1 motif. The number of peaks used for motif search was listed at the upper-right corner of each motif. (**D**) A motif footprint plot for the Ftz-f1-binding motif in replicate 2. (**E**) *Figure 4 continued on next page*

*Figure 4 continued*

An IGV plot shows the narrow peaks and motif sites in the gene region of *sim*. The motif sequences are shown in the magnified area. (**F**) The quantification of individual *sim* transcript expression in control and *ftz-f1*$^{RNAi1}$ egg chambers. The transcript expression is mined from RNA-seq data. The online version of this article includes the following figure supplement(s) for figure 4:

**Figure supplement 1.** Analysis of CUT&RUN narrow peaks.

identified at the proximal promoter region (−200 bp) of one of *sim*'s transcripts (FBtr0334613; *Figure 4E*). Most strikingly, FBtr0334613 was the only *sim* transcript expressed in stage-10B–12 follicles and was downregulated in *ftz-f1*–knockdown follicles, through reanalyzing the RNA-seq data using the HISAT-Stringtie (*Figure 4F*).

To test whether *sim* is indeed a downstream target of Ftz-f1, we performed Sim antibody staining in wild-type follicles and follicles with *ftz-f1* mutant clones. Sim was not expressed in follicle cells before stage 10B (except in stalk follicle cells connecting two egg chambers; *Figure 5A* and *Figure 5—figure supplement 1*). Sim was drastically upregulated in stage-10B/11 follicle cells (except anterior stretch follicle cells) and progressively downregulated to the lowest point at stage 13 (*Figure 5B–E*). Sim was re-upregulated at stage 14 and its function at this stage will be reported in another manuscript (*Figure 5F*). Consistent with the idea that *sim* is a downstream target of Ftz-f1, *ftz-f1* mutant follicle cells completely lack Sim expression at stage 10B and 12 (*Figure 5G–H*). In

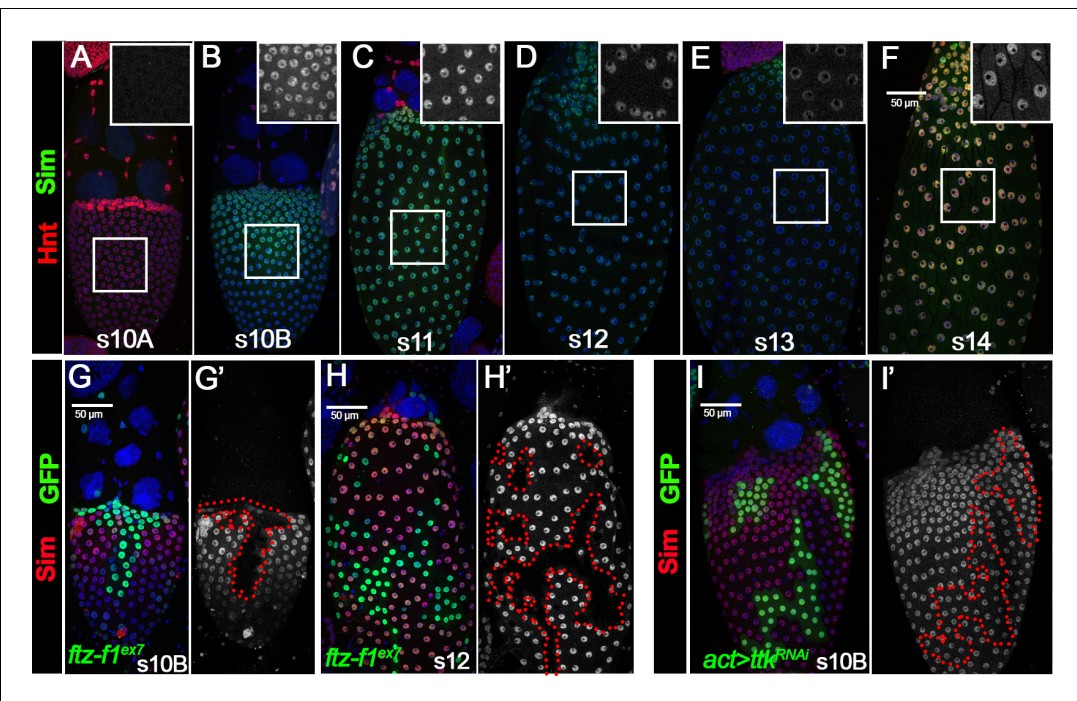

**Figure 5.** Ftz-f1 promotes Sim expression in stage-10B follicle cells. (**A–F**) The expression of Sim protein in late oogenesis. Sim protein is detected by anti-Sim antibody shown in green. Hnt expression (shown in red) is used to mark stage-10A (**A**) and stage-14 (**F**) follicles. The insets are higher magnification of Sim expression in squared areas. All images from A-F are acquired using the same microscopic settings. (**G–H**) Sim expression (red in G,H and white in G',H') in stage-10B (**G**) and stage-12 (**H**) egg chambers with *ftzf1*$^{ex7}$ mutant clones (marked by green GFP and outlined by dashed lines). (**I**) Sim expression (red in I and white in I') in stage-10B egg chambers with flip-out Gal4 clones (marked by green GFP and outlined by dashed line) overexpressing *ttk*$^{RNAi}$. Nuclei are marked by DAPI in blue.

The online version of this article includes the following figure supplement(s) for figure 5:

**Figure supplement 1.** Sim is detected in stalk follicle cells.
**Figure supplement 2.** Overexpression of *ftz-f1* induces premature Sim expression at stage 10A and disrupts follicle cell transition into stage 10B.

contrast, *ttk*-knockdown follicle cells have normal expression of Sim (*Figure 5I*). In addition, misexpression of *ftz-f1* is sufficient to induce premature Sim expression in stage-10A follicle cells (*Figure 5—figure supplement 2A–D*), which seemed to disrupt the follicle cell transition from stage 10A to stage 10B manifested by the continuous expression of Hnt and no upregulation of Cut at/after stage 10B (*Figure 5—figure supplement 2E–H*). Altogether, these data suggest that Sim is a direct target of Ftz-f1 but not Ttk69.

## Sim is required for follicle cell differentiation

To determine whether Sim is required for follicle cell differentiation, we generated flip-out Gal4 clones with overexpression of *sim^RNAi*. Follicle cells with s*im^RNAi* overexpression have no detectable Sim expression at stage 10B, 12, or 14 (*Figure 1—figure supplement 3B–D*), indicating efficient knockdown. Similar to the *ftz-f1* mutant follicle cells, *sim^RNAi*-overexpressing follicle cells also failed to fully upregulate Hnt expression at stage 14 (*Figure 6A*), as well as downregulate Cut, Ttk69, EcR. A, EcR.B1, and Br-C (*Figure 6B–F*). In addition, occasional faint expression of Hnt was detected in *sim*-knockdown follicle cells at stage 10B and 12 (*Figure 6G–H*), while the different level of Cut expression in *sim*-knockdown and adjacent wild-type follicle cells was detected at stage 10B but not at stage 12 (*Figure 6I–J*), similar to *ftz-f1* mutant follicle cells. The similarity between *ftz-f1* mutant and *sim*-knockdown follicle cells is not due to Sim regulating Ftz-f1 expression, as Ftz-f1 is properly upregulated in *sim*-knockdown follicle cells at stage 10B (*Figure 6K*). Our data suggest that Sim is essential for follicle cell differentiation in late oogenesis, like Ftz-f1.

We aimed to rescue differentiation defects of *ftz-f1*-knockdown follicle cells with misexpression of *sim* in the flip-out Gal4 system. Unfortunately, ectopic *sim* expression led to early follicle cell defects manifested by the smaller nuclei starting at stage 9, continuous expression of Hnt, and no expression of Cut from 7 to stage 14 (*Figure 6—figure supplement 1A–E*). Alternatively, we tested the ability of ectopic *sim* to rescue *ftz-f1* knockdown defects when driven by *Vm26Aa-Gal4*. However, ectopic expression of *sim* alone or in the *ftz-f1*-knockdown background led to disrupted Hnt and Cut expression patterns at stage 10B/11 (*Figure 6—figure supplement 2A–H*). These follicles showed mild rescue (if any) of Hnt, Cut, and Oamb-RFP expression at stage-14, but had abnormal morphology and no dorsal appendage formation as *ftz-f1*-knockdown follicles (*Figure 6—figure supplement 2I–P*). These data likely suggest that the level and temporal expression of Sim is essential for proper follicle cell differetiation. Nonetheless, the phenotypic similarity between *ftz-f1* and *sim* mutant follicle cells and the induction of *sim* expression by Ftz-f1 support the idea that Sim acts downstream of Ftz-f1 to promote follicle cell differentiation.

## Mouse SF-1 is sufficient to replace Ftz-f1's role in follicle cell maturation

Next, we examined whether ectopic expression of *ftz-f1* is sufficient to rescue *ftz-f1^RNAi* defects. As expected, flip-out Gal4 clones with both *ftz-f1* and *ftz-f1^RNAi2* showed rescue of Ftz-f1 expression in stage-10B follicle cells, despite it is slightly weaker than that in wild-type follicle cells (*Figure 7—figure supplement 1A–B*). This is likely due to *ftz-f1^RNAi* targeting not only endogenous *ftz-f1* gene but also ectopically expressed *ftz-f1* mRNA. We also observed complete rescue of Hnt and Cut expression (*Figure 7—figure supplement 1E–H*). Unlike *ftz-f1* overexpression alone (*Figure 5—figure supplement 2C*), we did not observe premature induction of Sim (*Figure 7—figure supplement 1C–D*), since Ftz-f1 was not overexpressed in early stages (*Figure 7—figure supplement 1A*).

To determine whether Ftz-f1's role in follicle cell differentiation is conserved, we investigated the potential of mouse SF-1 (mSF-1), the mouse homolog of Ftz-f1, to substitute for Ftz-f1 in follicle cell maturation. We generated flip-out Gal4 clones that express either *ftz-f1^RNAi2*, *mSF-1*, or both and examined follicle cell maturation markers. Consistent with *ftz-f1* mutant follicle cells (*Figure 3*), *ftz-f1^RNAi2*-overexpressing follicle cells could not upregulate Hnt expression at stage 14 (*Figure 7A*). In contrast, follicle cells with both *ftz-f1^RNAi2* and *mSF-1* had normal Hnt upregulation at stage 14, the same as follicle cells with *mSF-1* alone (*Figure 7B–D*). In addition, follicle cells with *ftz-f1^RNAi2* showed strong Cut expression at stage 14, while follicle cells with both *ftz-f1^RNAi2* and *mSF-1* had no Cut expression, similar to follicle cells with *mSF-1* alone (*Figure 7E–H*). These data suggest that mSF-1 can replace Ftz-f1's role in promoting follicle cell differentiation and maturation.

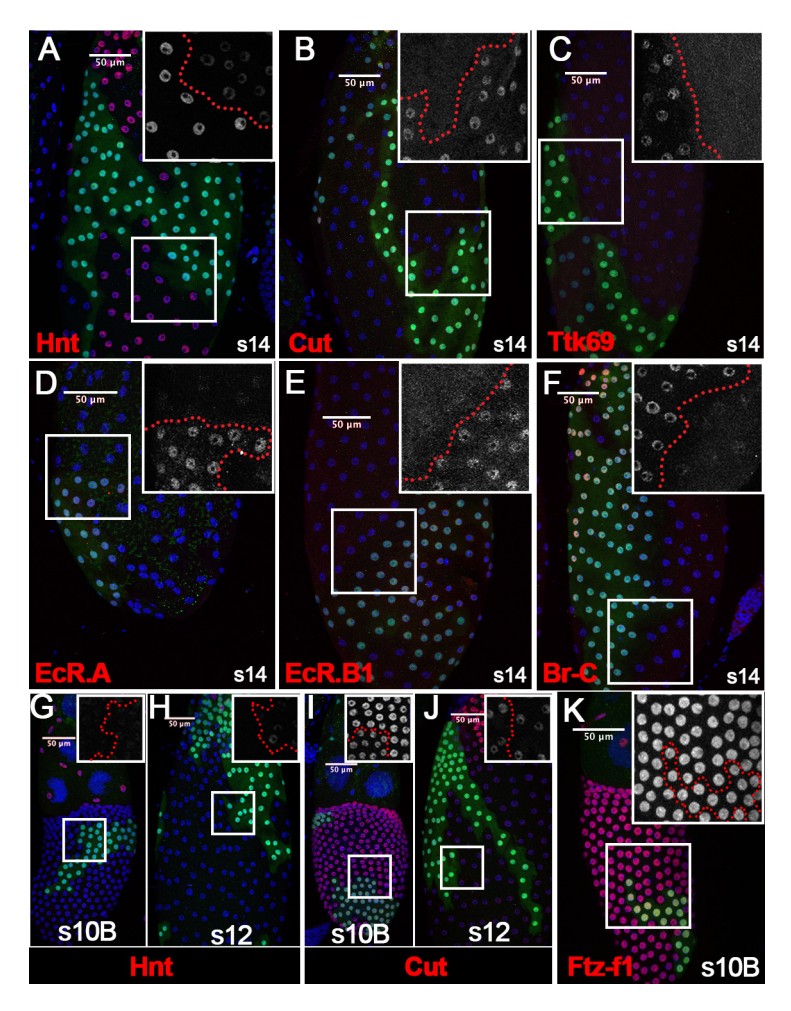

**Figure 6.** Sim promotes follicle cell differentiation into the final maturation stage. (A–F) The expression of Hnt (A), Cut (B), Ttk69 (C), EcR.A (D), EcR.B1 (E), and Br-C (F) in stage-14 egg chambers with flip-out Gal4 clones overexpressing *sim^{RNAi}* (marked by green GFP). Insets show higher magnification of Hnt (A), Cut (B), Ttk69 (C), EcR.A (D), EcR.B1 (E), and Br-C (F) expression in squared areas with the clone boundary marked by red dashed line. (G–H) Hnt expression (red in G-H) in stage-10B (G) and stage-12 (H) egg chambers with flip-out Gal4 clones overexpressing *sim^{RNAi}*. Insets show higher magnification of Hnt expression in squared areas with the clone boundary marked by red dashed line. (I–J) Cut expression (red in I-J) in stage-10B (I) and stage-12 (J) egg chambers with flip-out Gal4 clones overexpressing *sim^{RNAi}*. Insets show higher magnification of Cut expression. (K) Ftz-f1 expression (red in K) in stage-10B egg chambers with flip-out Gal4 clones overexpressing *sim^{RNAi}*. Insets show higher magnification of Ftz-f1 expression in squared areas with the clone boundary marked by red dashed line. Nuclei are marked by DAPI in blue.

The online version of this article includes the following figure supplement(s) for figure 6:

**Figure supplement 1.** Overexpression of *sim* disrupts earlier follicle cell differentiation.

**Figure supplement 2.** Rescue *ftz-f1* defect with *Vm26Aa-Gal4* driving *sim* overexpression.

Strikingly, we also noticed that ectopic mSF-1 was sufficient to promote premature differentiation of follicle cells. In wild-type follicle cells, Hnt expression was not downregulated until stage 10B; however, Hnt was prematurely downregulated in follicle cells with both *mSF-1* and *ftz-f1^{RNAi2}* at stage 10A but not in earlier stages (*Figure 7I–J*). In addition, Hnt was not re-upregulated until stage 14 in wild-type follicle cells but was prematurely upregulated in follicle cells with both *mSF-1* and *ftz-f1^{RNAi2}* at stages 12/13 (*Figure 7K–L*). In accordance with Hnt, Cut was prematurely upregulated in follicle cells with both *mSF-1* and *ftz-f1^{RNAi2}* at stage 10A and prematurely downregulated at stage 12/13 (*Figure 7M–P*). We consistently observed cytoplasmic staining of Cut in the clone cells,

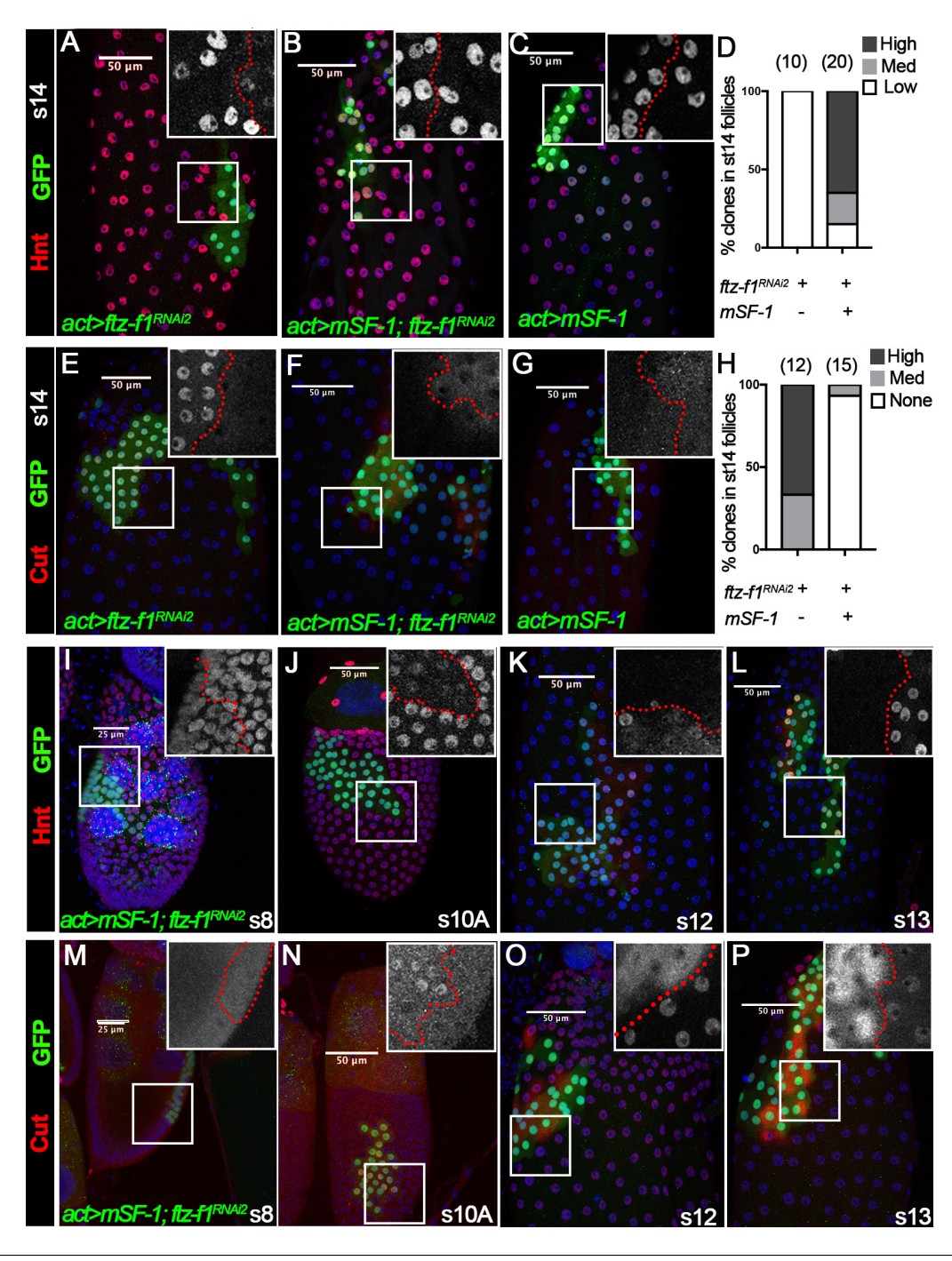

**Figure 7.** The role of Ftz-f1 in follicle cell maturation can be replaced by mSF-1. (A–H) Hnt expression (red in A-C) and Cut expression (red in E-G) in stage-14 egg chambers with flip-out Gal4 clones (marked by green GFP) overexpressing *ftz-f1^RNAi2* (A,E), *mSF-1;ftz-f1^RNAi2*(B,F), or *mSF-1* (C,G). The insets show higher magnification of Hnt expression (A–C) and Cut expression (E–G) in squared areas with the clone boundary marked by red dashed line. Quantification of clone phenotype is show in D for Hnt expression and H for Cut expression. The number of clones analyzed is noted above each bar. (I–P) Hnt expression (red in I-L) and Cut expression (red in M-P) in stage-8 (I and M), stage-10A (J and N), stage-12 (K, and O) and stage-13 (L and P) egg chambers with flip-out Gal4 clones overexpressing *mSF-1;ftz-f1^RNAi2* (marked by green GFP). Insets show the higher magnification of Hnt expression (I–L) and Cut expression (M–P) in squared areas with the clone boundary marked by red dashed line. Nuclei are marked by DAPI in blue.

*Figure 7 continued on next page*

*Figure 7 continued*

The online version of this article includes the following figure supplement(s) for figure 7:

**Figure supplement 1.** Ftz-f1 can rescue differentiation defects in *ftz-f1*–knockdown follicle cells.

indicating that Cut was evicted from follicle cell nuclei for degradation (*Figure 7O–P*). These data indicate that overexpression of *mSF-1* is sufficient to promote follicle cell differentiation prematurely.

## Mouse SF-1 is sufficient to induce Sim expression in the absence of Ftz-f1

The rescue of follicle cell maturation by mSF-1 prompted us to examine whether mSF-1 is also sufficient to restore Sim expression in *ftz-f1*–knockdown follicle cells. Like *ftz-f1* mutant clones (*Figure 5G–H*), Sim is barely detected in follicle cells with *ftz-f1^RNAi2^* overexpression at stage 14; however, it is readily detected in follicle cells with both *ftz-f1^RNAi2^* and *mSF-1* or *mSF-1* alone (*Figure 8A–D*). Most strikingly, ectopic *mSF-1* was able to prematurely induce Sim expression in follicle cells with *ftz-f1^RNAi2^* at stage 10A but not earlier stages (*Figure 8E–F*). In addition, Sim was also prematurely downregulated in these follicle cells at stage 12 (*Figure 8G–H*). All these data are consistent with the idea that ectopic *mSF-1* promotes the premature differentiation of follicle cells via Sim. In conclusion, our data suggest that ecdysone-induced Ftz-f1 promotes follicle cell differentiation and maturation partly via bHLH/PAS transcription factor Sim, and this role is likely conserved (*Figure 8I*).

## Discussion

### Transient regulation of Ftz-f1 in adult ovarian follicle cells by ecdysteroid signaling

Since the identification of the *ftz-f1* gene almost three decades ago (*Lavorgna et al., 1991*; *Ueda et al., 1990*), previous work has primarily focused on Ftz-f1's role in embryogenesis, larval development, pupation, and metamorphosis. The expression and function of Ftz-f1 in adult flies, particularly in oogenesis, is largely lacking. Work in this study demonstrated for the first time that Ftz-f1 is transiently expressed in the adult ovarian follicle cells from stage 10B to stage 12 according to three different reporters. It is worth noting that we were unable to detect Ftz-f1 expression in follicle cells before stage 10B, unlike the work reported previously (*Talamillo et al., 2013*). In addition, we didn't observe any morphological and molecular defects in *ftz-f1* mutant follicle cells before stage 10 (data not shown).

Ftz-f1 antibody used in this study is raised against βFtz-f1 protein; however, it can potentially detect αFtz-f1 since αFtz-f1 and βFtz-f1 share common C-terminal regions (personal communication with Dr. Ueda). Therefore, it is unknown whether follicular Ftz-f1 is αFtz-f1 or βFtz-f1. Since αFtz-f1 is maternally supplied and only detected in early embryos, we favor the idea that it is βFtz-f1 expressed in follicle cells. This is consistent with the fact that follicular Ftz-f1 is regulated by ecdysteroid signaling, similar to the transient expression of βFtz-f1 after each ecdysone pulses during larval and pupal development (*Yamada et al., 2000*).

It is striking that follicular Ftz-f1 is so transiently expressed, similar to transient expression of βFtz-f1 in development. Our data showed that ecdysteroid signaling is essential for Ftz-f1 expression at stage 10B. It seems contradictory to the fact that βFtz-f1 is inhibited by high ecdysone titer and only induced when ecdysone titer is low during development (*Broadus et al., 1999*; *Woodard et al., 1994*; *Yamada et al., 2000*). However, there's no precise measurement of ecdysone titer at each stage of oogenesis. It is plausible that ecdysone signaling at stage 10A leads to sequential activation of genes that are responsible for Ftz-f1 expression at stage 10B. Consistent with this idea, *Cyp18a1*, encoding a cytochrome P450 enzyme involved in inactivating 20-hydroxyecdysone and inducing Ftz-f1 expression during the prepupa-to-pupa transition (*Rewitz et al., 2010*), is significantly enriched in stage-10B follicles and likely required for follicle cell differentiation (*Tootle et al., 2011*). Unfortunately, either overexpression or knockdown of *Cyp18a1* did not affect Ftz-f1 expression in follicle

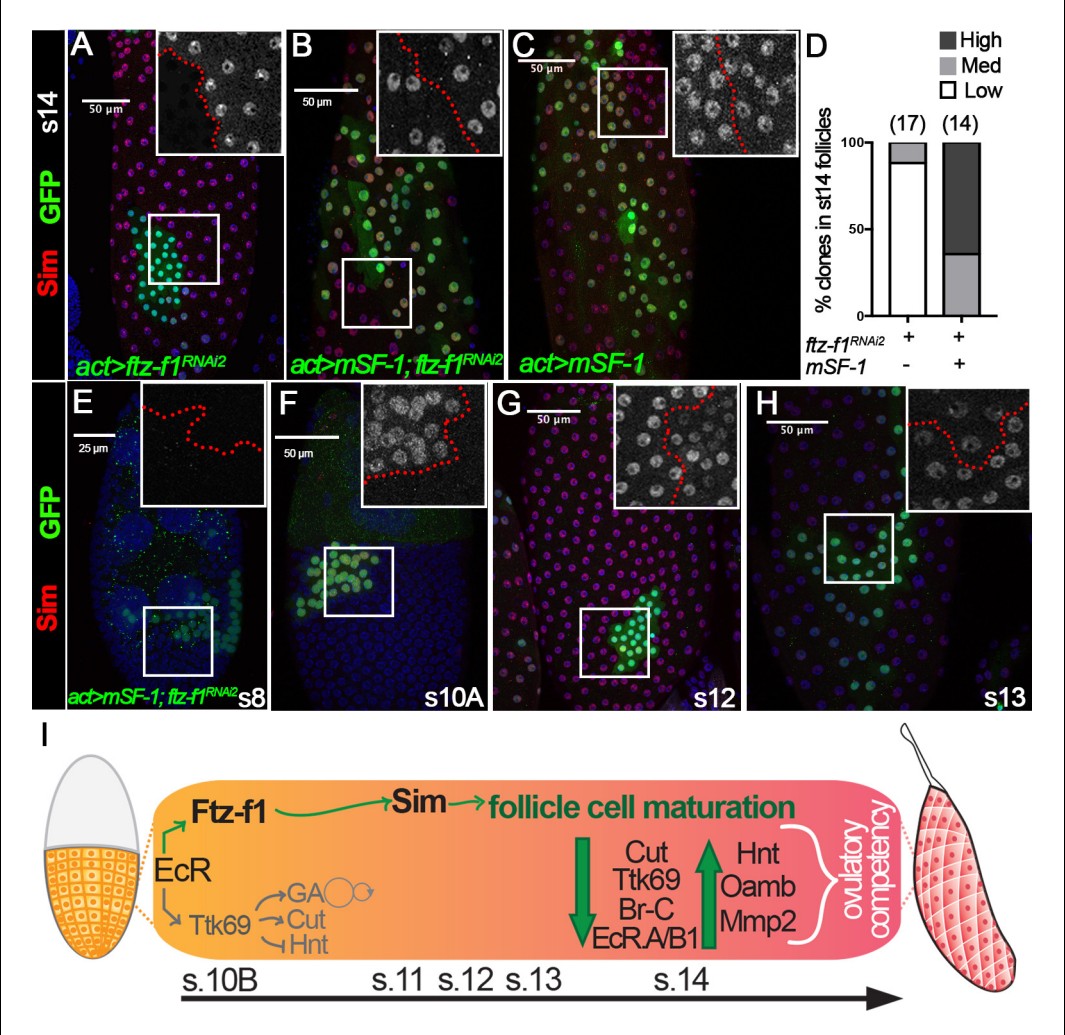

**Figure 8.** Sim expression can be rescued by mSF-1. (A–D) Sim expression (red in A-C) in stage-14 egg chambers with flip-out Gal4 clones (marked by green GFP) overexpressing *ftz-f1^RNAi2* (A), *mSF-1;ftz-f1^RNAi2*(B), or *mSF-1* (C). Insets show higher magnification of Sim expression (A–C) in squared areas with the clone boundary marked by red dashed line. Quantification of Sim expression in these clones is shown in D. The number of clones analyzed is noted above each bar. (E–H) Sim expression (red in E-H) in stage-8 (E), stage-10A (F), stage-12 (G), and stage-13 (H) egg chambers with flip-out Gal4 clones (marked by green GFP in E-H) overexpressing *mSF-1;ftz-f1^RNAi2*. Insets show higher magnification of Sim expression (E–H) in squared areas with the clone boundary marked by red dashed line. Nuclei are marked by DAPI in blue. (I) A schematic drawing shows the role of Ftz-f1 and Sim in follicle cell differentiation in late oogenesis. At stage-10B Ftz-f1 expression is required for induction of Sim. Expression of Cut, Ttk69, Br-C, and EcR.A/B1 are high in stage-10B follicle cells and downregulate by stage-13. Expression of Hnt, Oamb, and Mmp2 are absent in stages 10B-13, and are then robustly upregulated in stage-14 follicle cells. GA: gene amplification.

cells. In addition, exogenous 20E is also not sufficient to induce Ftz-f1 expression in earlier stages. Thus, Ftz-f1 expression is precisely regulated in follicle cells and is not sensitive to the 20E level. It will be interesting to investigate whether other ecdysone-induced genes that regulate βFtz-f1 expression during the larva-to-pupa transition, such as Blimp-1, DHR3, E75, and Nos (*Akagi et al., 2016*; *Cáceres et al., 2011*; *Yamanaka and O'Connor, 2011*), contribute to precise upregulation of Ftz-f1 in stage-10B follicle cells. It is unknown what factors contribute to downregulation of Ftz-f1 at stage 12. It is worth noting that several Ftz-f1-binding sites were identified in the *ftz-f1* gene (*Supplementary file 2*) and that βFtz-f1 can negatively regulates its own expression during prepupa-

to-pupa transition (*Woodard et al., 1994*). A similar negative-feedback mechanism could occur in follicle cells.

## Ftz-f1 functions as a competency factor for follicle cells to progress into final maturation

Previous work regarding follicular epithelium mostly focused on egg chambers —before stage 10, at the stage 10A/10B transition, or at the stage 13/14 transition (*Duhart et al., 2017*; *Klusza and Deng, 2011*; *Knapp et al., 2019*; *Osterfield et al., 2017*). Little is known about how stage-10B follicle cells differentiate into final maturation. With both global knockdown and mutant clone analyses, our work clearly demonstrated that Ftz-f1 is a key factor required for promoting stage-10B follicle cells to differentiate into final maturation, which is essential for releasing fertilizable oocytes at the end of oogenesis. Molecular marker analysis showed that all known stage-14 follicle cell markers, including upregulated Hnt, Oamb, Mmp2 expression and downregulated Cut, Ttk69, Br-C, EcR.A/B1 expression (*Figure 8I*), are disrupted in *ftz-f1* mutant follicle cells. In fact, *ftz-f1* mutant follicle cells seem to be arrested at the end of stage 10B. All these data suggest that Ftz-f1 is a master regulator for the final differentiation of follicle cells after stage 10B. Consistent with this idea, loss of *ftz-f1* also led to disrupted dorsal appendage formation and chorion gene amplification, and likely eggshell formation.

It is not completely understood how Ftz-f1 can have such profound influence on cell differentiation. During the larva-to-pupa transition, Ftz-f1 seems to regulate ecdysteroid synthesis enzymes and thus influence the next ecdysone pulse (*Akagi et al., 2016*; *Parvy et al., 2005*). Could the same mechanism apply in follicle cells? Indeed, we have previously demonstrated that another pulse of ecdysteroid signaling occurs in stage-14 follicle cells in addition to the ecdysteroid signaling at the stage 10A/10B transition (*Knapp and Sun, 2017*). This is controlled by the upregulation of Shade (Shd), the enzyme converting ecdysone to active 20-hydroxyecdysone. However, preliminary analysis showed that Shd is continuously expressed in *ftz-f1* mutant follicle cells (data not shown). Therefore, Ftz-f1 is unlikely to regulate follicle cell differentiation through regulating the next pulse of ecdysteroid signaling. This is also supported by the fact that Ftz-f1 promotes follicle cell differentiation in a cell-autonomous fashion and that Sim functions as a downstream target to promote follicle cell differentiation (see below).

Few direct targets of Ftz-f1 have been identified. Among those, *ftz*, *Edg84A*, and *Adh* are best characterized, and all of them have Ftz-f1 binding motif (YCAAGGYCR) in the promoter region within 500 bp upstream of TSS (*Ayer and Benyajati, 1992*; *Murata et al., 1996*; *Ueda et al., 1990*). With RNA-seq, we identified 389 differentially expressed genes in follicles with *ftz-f1* knockdown. GO term analysis showed that genes related to intrinsic and integral components of membrane are most enriched among downregulated genes, while genes related to secondary active transmembrane transporter activity, developmental process, and chorion are most enriched among upregulated genes (*Supplementary file 1*). Using CUT&RUN experiment, we identified Ftz-f1 binding motifs in follicle cells that were similar to the canonical Ftz-f1 binding motif (YCAAGGYCR). More than 250 sites could be potential Ftz-f1 direct binding sites (*Supplementary file 2*). Combining both experiments, we were able to identify 15 genes/transcripts that could be potential direct targets of Ftz-f1 in follicle cells. Among these, our data illustrated that one of *sim*'s transcript (FBtr0334613) is the only transcript expressed in follicle cells and is the direct target of Ftz-f1 (*Figure 4*). This is also supported by our finding that *sim3.7-Gal4*, which utilizes a 3.7 kb promoter region of *sim*'s longest transcript (FBtr0082711) that does not contain Ftz-f1-binding site (*Xiao et al., 1996*), was not detected in follicle cells (data not shown). In the future, it will be interesting to isolate the entire promoter region that is required for *sim* expression in follicle cells and identify all the factors regulating its expression. In addition, future work will be focused on the other direct targets of Ftz-f1 to better understand the molecular network of Ftz-f1 regulated follicle cell differentiation and maturation.

## The transcription factor Sim functions as a novel target of Ftz-f1 for follicle cell differentiation

Sim is a master regulator of central nervous system (CNS) midline cell development and has been extensively studied in the development of the CNS midline, the central complex, and optic ganglia in the last two decades (*Nambu et al., 1991*; *Pielage et al., 2002*; *Umetsu et al., 2006*). Its role

outside the nervous system is sparse. Our findings here also illustrated for the first time that Sim is upregulated in stage-10B follicle cells and is essential for follicle cell differentiation. This is consistent with a previous report that *sim* mutant flies are sterile (*Pielage et al., 2002*). We also demonstrated that Sim upregulation depends on Ftz-f1, not vice versa, which places Sim downstream of Ftz-f1. In addition, phenotypic defects of *sim*-knockdown follicle cells are strikingly similar to those of *ftz-f1* mutant follicle cells. Furthermore, mSF-1 overexpression leads to premature Sim upregulation at stage 10A as well as premature follicle cell differentiation. All these data support the conclusion that Sim function as the downstream effector of Ftz-f1 to promote follicle cell differentiation. Sim belongs to the bHLH/PAS transcription factor family and dimerizes with another bHLH-PAS transcription factor Tango to activate downstream gene expression (*Ohshiro and Saigo, 1997*; *Sonnenfeld et al., 1997*). It will be interesting to investigate whether Tango is a cofactor for Sim in follicle cells and what are the direct targets of Sim in follicle cell differentiation in the future. It will be also interesting to know whether Sim also acts downstream of Ftz-f1 during larval and pupal development.

Our work also illustrated the importance of precise control of Sim expression in follicle cells. Ectopic *sim* expression in early-stage follicle cells seemed to disrupt the organization of the follicle-cell monolayer (*Figure 6—figure supplement 1*). It also disrupts the endoreplication as follicle cell nuclei are smaller than the adjacent wild-type cells. This is not due to the disruption of Notch signaling and mitotic/endocycle transition (*Sun and Deng, 2005*; *Sun and Deng, 2007*), because Cut is properly downregulated in these cells and the nuclei size defect is only manifested after stage 8. Therefore, premature upregulation of Sim may also disrupt the cell differentiation program. In addition, Sim is also expressed in stalk follicle cells and its role in stalk follicle cells is completely unknown.

## Conservation of NR5A nuclear receptor signaling in ovarian follicle cells

The mammalian NR5A homolog SF-1, is expressed in somatic follicle cells of the ovary in both rodents and humans (*Hinshelwood et al., 2003*; *Tajima et al., 2003*), and loss of this SF-1 expression in murine granulosa cells leads to a severe depletion of developing follicles and infertility (*Pelusi et al., 2008*). Despite the critical role for SF-1 in female fertility, it still remains unknown how SF-1 within these follicle cells regulates folliculogenesis. *Drosophila* poses as a valuable model for the study of the function of NR5A receptors, considering the DNA binding sequence of NR5A receptors is highly conserved from *Drosophila* to humans, with over 80% in sequence similarity (*Fayard et al., 2004*). Furthermore, studies have already begun to show the functional conservation of these NR5A receptors in both the embryo and female reproductive tract of *Drosophila* (*Lu et al., 2013*; *Sun and Spradling, 2012*; *Splinter et al., 2018*). In this work, we demonstrated that Ftz-f1 is also expressed in the somatic follicle cells of the ovary and plays a crucial role in female fertility, akin to SF-1. Furthermore, our work demonstrated that Ftz-f1's function in follicle cell differentiation is functionally conserved, as mSF-1 is sufficient to rescue defects in follicle cell maturation caused by loss of Ftz-f1. Our results also showed that mSF-1 is sufficient to induce expression of the Ftz-f1 target Sim. The mammalian homologs of Sim are encoded by *sim1* and *sim2* (*Yamaki et al., 1996*). The role of Sim1 and Sim2 in female fertility have never been studied; however, Sim2 seems to be expressed in human ovarian follicle cells (according to Human Protein Atlas). Thus, it would be interesting to probe if Sim1 or Sim2 is expressed in ovarian follicle cells and whether they function downstream of SF-1 for follicle development. Overall, our findings could help to further elucidate the genetic and molecular mechanisms of NR5A signaling and how it regulates follicle development and female fertility.

## Materials and methods

### *Drosophila* genetics and clone induction

Flies were reared on standard cornmeal and molasses food at 25°C, unless noted otherwise. *ftz-f1^{ex7}* is a P-element excision line and is considered as a null allele (*Yamada et al., 2000*). For *ftz-f1* expression analysis, *ftz-f1::GFP.FLAG* [Bloomington *Drosophila* Stock Center (BDSC), stock #38645] and *ftz-f1^{fs(3)2877}* (*Karpen and Spradling, 1992*) were utilized. The protein trap line *Mmp2::GFP/Cyo* (*Deady et al., 2015*) was used for Mmp2 expression. The *Vm26Aa-Gal4* (*Peters et al., 2013*) was used to drive expression in follicle cells starting at stage 10. Isolation and identification of stage-14

follicles for follicle rupture assay were performed using *Oamb-RFP* (*Knapp et al., 2019*) or *47A04-LexA* (BDSC, stock #54873) driving *lexAop2-6XGFP* (BDSC, stock #52265). *sim3.7-Gal4* (*Xiao et al., 1996*) was also from BDSC (stock #26784). The following transgenic lines were used to knock down or overexpress genes in experiments: *UAS-EcR^DN* (BDSC, stock #6872), *UAS-ttk^RNAi* [Vienna *Drosophila* Resource Center (VDRC), stock #101980], *UAS-Cyp18a1* (*Rewitz et al., 2010*), *UAS-Cyp18a1^RNAi* (VDRC, stock #5602), *UAS-ftz-f1^RNAi1* (BDSC, stock #33625), *UAS- ftz-f1^RNAi2* (VDRC, stock #104463), *UAS-ftz-f1* (*Yussa et al., 2001*), *UAS-sim^RNAi* (VDRC, stock #26888), UAS-sim-3xHA (Fly-ORF, stock #000719) and *UAS-mSF1* (*Yussa et al., 2001*). Ecdysone sensor *hsGal4^DBD-EcR^LBD*, *UAS-nlacZ* was a gift by Wu-Min Deng (*Kozlova and Thummel, 2002*). All experiments involving RNAi lines are performed at 29°C and contain *UAS-dcr2* in order to enhance the RNAi efficiency. Control flies for all experiments were prepared by crossing Gal4 driver to Oregon-R flies.

Mosaic analysis with repressible cell marker (MARCM) was used to generate follicle cell clones homozygous for the *ftz-f1^ex7* allele, via crossing *hsFLP, tub-Gal4, UAS-GFP; tub-Gal80, FRT2A/TM6B* to *ftz-f1^ex7, FRT 79D/TM3, Ser*. Flip-out Gal4 clones were generated using either the *hsFLP; act <CD2<Gal4, UAS-GFP/TM3* or *hsFLP; act <CD2<Gal4, UAS-RFP/TM3* stock to cross to indicated transgenes of interest. For clone induction, adult female progeny with correct genotypes were heat shocked for 45 min at 37°C to induce FLP/FRT mediated recombination and incubated at 25°C for 2–4 days before dissection. For analysis of EcR ligand sensor, flies were heat shocked for 45 min at 37°C and allowed to recover at 29°C for 16 hr before dissection. Dissected ovaries were treated with 100 nM of 20E (Cayman Chemical) in Grace's medium for five hours before fixation and antibody staining.

## Ovulation assays

Egg-laying experiments were performed as previously described (*Deady and Sun, 2015*). Five-day-old females (fed with wet yeast for 1 day) were housed with Oregon-R males (five females: 10 males) in one bottle to lay eggs on molasses plates over two days at 29°C (with removal and replacement of plates every 22 hr). After egg laying, the ovary pairs for each female were dissected out and the number of mature follicles within the ovary pair were quantified.

The ex vivo follicle rupture assays were performed as described previously (*Knapp et al., 2018*). Ovaries from 5- to 6-day-old virgin females fed with wet yeast for 3 days were dissected out and stage-14 follicles were isolated in Grace's insect medium (Caisson Labs, Smithfield, UT). After isolation, follicles were separated into groups ~ 30 follicles and cultured at 29°C for 3 hr in culture medium (Grace's insect medium +10% fetal bovine serum + 1% penicillin-streptomycin) containing 20 μM OA (Sigma-Aldrich) or 2 μM ionomycin (Cayman Chemical, Ann Arbor, MI). Each data point represents the percentage (mean ± standard deviation (SD)) of ruptured follicles per experimental group.

## Superoxide detection

Measurement of superoxide production was performed as previously described (*Li et al., 2018*), with slight modifications. Five mature follicles were isolated and placed in each well of a 96-well plate with 100 μl of Grace's insect medium containing either 20 μM OA or 2 μM ionomycin and 200 μM of L-012 (Wako Chemicals). Plates were placed in a CLARIOstar microplate reader (BMG Labtech) for luminescence reading for 60 min. Eight to ten wells (technical repeats) were used in each experiment for each genotype, and the mean ± standard error of the mean (SEM) of the technical repeats was calculated. Each experiment was performed at least twice.

## Immunostaining, EdU detection, and microscopy

Immunostaining was performed following a standard procedure (*Sun and Spradling, 2012*). The following primary antibodies were used: mouse anti-Hnt (1G9, 1:75), anti-Cut (2B10, 1:15), anti-Br-C (25E9.D7, 1:15), anti-EcR.A (15G1a, 1:30), and anti-EcR.B1 (AD4.4, 1:30) from the Developmental Study Hybridoma Bank; rabbit anti-Ftz-f1 (1:50000; a gift from Dr. Hitoshi Ueda, Okayama University, Japan), rabbit anti-Ttk69 (1:100; a gift from Dr. Wanzhong Ge, Zhejiang University, China), rabbit anti-GFP (1:4000; Invitrogen), mouse anti-GFP (1:1000; Invitrogen), rabbit anti-RFP (1:2000, MBL international), Chicken anti-β-Gal (ab9361, 1:500; Abcam), and guinea pig anti-Sim (1:1000; a gift from Dr. Stephen Crews, University of North Carolina at Chapel Hill School of Medicine, Chapel Hill,

USA). Alexa Fluor 488 and Alexa Flour 568 goat secondary antibody (1:1000; Invitrogen) were used as secondary antibodies.

EdU detection was performed as previously described (*Alexander et al., 2015*). Ovaries were dissected out in room temperature Grace's insect medium and incubated in 50 µM EdU for 30 min. Ovaries were then fixed in 4% EM-grade paraformaldehyde for 13 min and permeabilized in PBX (0.1% TritonX in PBS) for 30 min. For detection of EdU, the Invitrogen's Click-iT EdU Alexa Fluor 555 Imaging Kit was utilized following the manufacturer's instructions.

Images were acquired using a Leica TCS SP8 confocal microscope or Leica MZ10F fluorescent stereoscope with a sCOMS camera (PCO.Edge) and assembled using Photoshop software (Adobe) and ImageJ.

## RNA-Seq and data analysis

Around 60 stage-10B–12 egg chambers from 7 to 10 flies were isolated in Grace's medium (Caisson labs) and grounded in 300 µl of TRIzol (Life Technologies, 15596018) directly. Total RNAs were extracted using Direct-zol RNA MicroPrep Kit (Zymo Research, Irvine, CA). mRNA libraries were prepared using Illumina TruSeq Stranded mRNA Sample Preparation kit following the manufacturer's protocol (Illumina, San Diego, CA) and were then sequenced on an Illumina NextSeq 550 sequencer to achieve single-end 75 bp reads in UConn's Center for Genome Innovation. Three biological replicates were prepared for each genotype.

Raw reads from RNA-seq were trimmed with Sickle (-q 30 l 50). Trimmed reads were mapped to *Drosophila melanogaster* genome (dm6) with HISAT2 (*Kim et al., 2015*). The counts were generated against the features with HTSeq-count (*Anders et al., 2015*). Principal component analysis (PCA) was used to test the reproducibility between the replicates. One *ftz-f1*-knockdown sample was an outlier due to unknown reason and was dropped from the analysis. The differential expression of genes between conditions were evaluated using DESeq2 (*Love et al., 2014*). In DESeq2, genes showing less than 10 cumulative counts across the compared samples were dropped from the analysis. Genes with (a) base mean counts >10, (b) a False discovery Rate (FDR) < 0.01, and (c) absolute value of log2FoldChange > 1 were considered to be significant and used in the downstream analysis. For transcript level expression, HISAT, Stringtie and Ballgown method was used (*Pertea et al., 2016*). Stringtie was used to estimate FPKM for each transcript.

## CUT&RUN and data analysis

The sample preparation for CUT&RUN followed the previous protocol with slight modification (*Skene et al., 2018*). In short, approximately 200 stage-10B–13 egg chambers from ~15 *ftz-f1::GFP. FLAG* females were isolated in 1xPBS. These egg chambers were equally separated into two halves, quickly spun and washed three times with wash buffer (20 mM HEPES-NaOH pH 7.5, 150 mM NaCl, 0.5 mM Spermidine, with 1x protease inhibitor EDTA free), and incubated in primary antibody at 4°C overnight. Samples were washed twice in dig-wash buffer and incubated for 1 hr at 4°C with protein-AG MNase (1:800) expressed and purified in house with the plasmid from Addgene (#123461). For chromatin digestion and release, high $Ca^{2+}$/low salt option was chosen and performed as in *Meers et al., 2019*. For library preparation, NEBNext Ultra II DNA Library Prep Kit (NEB) was performed as described in *Liu et al., 2018*. For amplification, after the addition of indexes, 14 cycles of 98°C, 20 s; 65°C, 10 s were run. A 1.2x SPRI bead cleanup was performed (Agencourt Ampure XP, Beckman). Libraries were sequenced on an Illumina NextSeq 500 sequencer to achieve pair-end 75 bp reads. The following primary antibody were used: mouse anti-FLAG M2 (1:250; Sigma F1804; experimental antibody) and mouse IgG1 (1:125; Sigma MABC002, control antibody). Three biological replicates were performed for each experimental antibody and control antibody.

For the data analysis, we followed the CUT&RUNTools workflow with the following modification (*Zhu et al., 2019*). In short, trimmed pair-end reads were mapped to *Drosophila melanogaster* genome (dm6) using Bowtie2 (option `–dovetail –local –very-sensitive-local –no-unal –no-mixed –no-discordant`) (*Langmead and Salzberg, 2012*). Fragments < 120 bp from experimental and control samples were used in MACS2 (*Zhang et al., 2008*) for identifying the narrow peaks (`macs2 callpeak -t experiment.bam -c control.bam -g dm -f BAMPE -n outprefix –outdir outdir -q 0.01 -B –SPMR –keep-dup all`). de novo motif search and motif footprint analysis were exactly followed in *Zhu et al., 2019*. Chipseeker was used to

analyze the peak distribution and motif sites relevant to nearest genes (*Yu et al., 2015*). All sequencing data are deposited in NCBI Sequence Read Archive (SRA) with BioProject ID PRJNA624186.

## Statistical analysis

Statistical tests were performed using Prism 7 (GraphPad, San Diego, CA).

Quantification results are presented as mean ± SD or mean ± SEM as indicated. Statistical analysis was conducted using Student's **t**-test.

## Acknowledgements

We thank Drs. Celeste Berg, Stephen Crews, Wu-Min Deng, Jean-Maurice Dura, Wanzhong Ge, Leslie Pick, Allan Spradling, Mike O'Connor and Hitoshi Ueda for sharing reagents and fly lines; Bloomington *Drosophila* Stock Center and Vienna *Drosophila* Resource Center for fly stocks; and Developmental Studies Hybridoma Bank for antibodies. Special thanks go to Drs. Steven Henikoff, Michael Meers, and Guo-Cheng Yuan for addressing questions related to CUT&RUN experiment. We thank Lylah Deady, Yuping Huang, and Rebecca Oramas in Sun's laboratory for technical support and discussion and UConn Center for Genome Innovation for providing sequencing service. We appreciate constructive comments from anonymous reviewers. The Leica SP8 confocal microscope is supported by an NIH Award (S10OD016435) to Akiko Nishiyama. JS is supported by the University of Connecticut Start-Up Fund, NIH/National Institute of Child Health and Human Development Grants (R01-HD086175 and R01-HD097206), and the Bill and Melinda Gates Foundation (Opp1160858 and Opp1203047).

## Additional information

### Funding

| Funder | Grant reference number | Author |
|---|---|---|
| Eunice Kennedy Shriver National Institute of Child Health and Human Development | R01-HD086175 | Jianjun Sun |
| Bill and Melinda Gates Foundation | Opp1160858 | Jianjun Sun |
| Bill and Melinda Gates Foundation | Opp1203047 | Jianjun Sun |
| Eunice Kennedy Shriver National Institute of Child Health and Human Development | R01-HD097206 | Jianjun Sun |

The funders had no role in study design, data collection and interpretation, or the decision to submit the work for publication.

### Author contributions

Elizabeth M Knapp, Data curation, Formal analysis, Investigation, Methodology, Writing - original draft; Wei Li, Methodology, Performed library preparation for RNA-seq and CUT&RUN; Vijender Singh, Methodology, facilitate the analysis of RNA-seq and CUT&RUN data; Jianjun Sun, Conceptualization, Supervision, Funding acquisition, Investigation, Methodology, Project administration, Writing - review and editing

### Author ORCIDs

Elizabeth M Knapp (ID) https://orcid.org/0000-0002-3524-9389
Wei Li (ID) http://orcid.org/0000-0001-8699-5325
Jianjun Sun (ID) https://orcid.org/0000-0002-6015-738X

### Decision letter and Author response

Decision letter https://doi.org/10.7554/eLife.54568.sa1

Author response https://doi.org/10.7554/eLife.54568.sa2

## Additional files

### Supplementary files

• Supplementary file 1. Table of differentially expressed genes.

• Supplementary file 2. Table of motif site analysis.

• Supplementary file 3. Table of direct targets of Ftz-f1.

• Supplementary file 4. Key Resources Table.

• Transparent reporting form

### Data availability

All data generated or analysed during this study are included in the manuscript and supporting files. Sequencing data have been deposited in SRA under BioProject ID PRJNA624186.

The following dataset was generated:

| Author(s) | Year | Dataset title | Dataset URL | Database and Identifier |
|---|---|---|---|---|
| Knapp EM, Li W, Singh V, Sun J | 2020 | Direct targets of Ftz-f1 in Drosophila follicle cells | http://www.ncbi.nlm.nih.gov/bioproject/?term=PRJNA624186 | NCBI BioProject, PRJNA624186 |

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
