## [Decision Letter]

**Acceptance summary:**

In this manuscript, Knapp and Sun provide evidence that the nuclear hormone receptor Ftz-F1 regulates follicle cell development and ovulation by controlling the expression of a second transcription factor Single-minded. The results provide insights into the transcriptional hierarchies that govern the late maturation of follicle cells as they prepare for ovulation.

**Decision letter after peer review:**

Thank you for submitting your article "NR5A nuclear receptor Ftz-f1 promotes follicle maturation and ovulation via bHLH/PAS transcription factor Single-minded" for consideration by *eLife*. Your article has been reviewed by three peer reviewers, including Michael Buszczak as the Reviewing Editor and Reviewer #1, and the evaluation has been overseen by Anna Akhmanova as the Senior Editor. The following individual involved in review of your submission has agreed to reveal their identity: Todd G Nystul (Reviewer #2).

The reviewers have discussed the reviews with one another and the Reviewing Editor has drafted this decision to help you prepare a revised submission.

Summary:

In this manuscript, Knapp and Sun provide evidence that the nuclear hormone receptor Ftz-F1 regulates follicle cell development and ovulation by controlling the expression of a second transcription factor Single-minded. The study starts with showing that Ftz-F1 expression is induced in stage 10B follicle cells through an ecdysone-dependent mechanism. Loss of Ftz-F1 in late stage follicle cells results in reduced egg laying, marked by retention of stage 14 follicles and a reduced response to octopamine. Further experiments showed that Ftz-F1 knockdown prevents the differentiation of follicle cells, resulting in the retention of Cut and Ttk69 expression and patterns of gene expression that indicates the *ftz-f1* mutant follicle cells remained arrested in Stage 10B. The authors found that knockdown of single-minded (*sim*) resulted in similar defects in follicle cells. Further work suggested that *sim* is a potential downstream target of ftz-f1. Finally, the authors show that transgenic expression of mSF1, the mammalian *ftz-f1* homolog, can rescue the *ftz-f1* knockdown phenotypes.

Essential revisions:

In general, the data are nicely presented and the experiments well-controlled. Showing evolutionary functional conservation between Ftz-f1 and mSF-1 is a clear strength of the paper. The results provide further insights into the transcriptional hierarchies that govern the late maturation of follicle cells, as they prepare for ovulation.

However, certain conclusions, particularly in regards to the relationship between Ftz-f1 and *sim*, are not fully supported by the data. For example, the authors state "These data suggest that Sim is a downstream target of Ftz-f1 but not Ttk69". Loss of Ftz-f1 clearly delays follicle cell maturation, and any loss/gain of gene expression could be due to this developmental arrest, not direct regulation.

1) At the very least, the authors should perform ChIP-seq and/or RNA-seq experiments to define the transcription targets of Ftz-f1 and/or Sim. Potential targets should be validated. Without these data, the paper is somewhat limited in scope, and all the changes in marker gene expression may simply be a reflection in the block in follicle cell differentiation.

2) Determining whether *sim* overexpression is able to rescue phenotypes caused by Ftz-f1 knockdown is a critical test of the idea that *sim* is genetically downstream of Ftz-f1. The authors attempted this experiment by simultaneously driving Ftz-f1^RNAi^ and *sim* with Act-Gal4 but the results were complicated because overexpression of *sim* caused a phenotype earlier in follicle cell differentiation. Thus, the authors should test whether they can circumvent this problem by using another Gal4 driver with a more limited expression pattern, like the *Vm26Aa-Gal4* used in Figure 2 or the *sim-Gal4* available at BDSC. There are other genetic tricks (Gal80ts etc.) that should allow for the expression of *sim* in stage 10B follicles when Ftz-f1 in knocked down. These experiments should be explored further and are absolutely critical for the conclusion described in the title.

3) The authors should provide additional data that the induction of *ftz-f1* expression is determined by ecdysone levels. For example, can Ftz-f1 be induced by exogenous ecdysone? They should also demonstrate the importance of this temporally restricted expression pattern by expressing Ftz-F1 prematurely and/or delaying its expression.

---

## [Author Response]

Essential revisions:In general, the data are nicely presented and the experiments well-controlled. Showing evolutionary functional conservation between Ftz-f1 and mSF-1 is a clear strength of the paper. The results provide further insights into the transcriptional hierarchies that govern the late maturation of follicle cells, as they prepare for ovulation.However, certain conclusions, particularly in regards to the relationship between Ftz-f1 and sim, are not fully supported by the data. For example, the authors state "These data suggest that Sim is a downstream target of Ftz-f1 but not Ttk69". Loss of Ftz-f1 clearly delays follicle cell maturation, and any loss/gain of gene expression could be due to this developmental arrest, not direct regulation.

Indeed, we never claimed that *sim* is a direct target of Ftz-f1 in the previous version of our manuscript as we were concerned that the gene regulation could be due to developmental arrest. We appreciate the criticisms and constructive comments reviewers raised. With the additional RNA-seq and CUT&RUN experiments performed, we were able to determine that *sim* is a direct target of Ftz-f1 in follicle cells (see detailed explanation in the next question).

1) At the very least, the authors should perform ChIP-seq and/or RNA-seq experiments to define the transcription targets of Ftz-f1 and/or Sim. Potential targets should be validated. Without these data, the paper is somewhat limited in scope, and all the changes in marker gene expression may simply be a reflection in the block in follicle cell differentiation.

As reviewers suggested, we performed both RNA-seq and CUT&RUN experiments to define the direct targets of Ftz-f1. By comparing the expression profile of stage-10B-12 follicles in control and *ftz-f1^RNAi1^* females with *Vm26Aa-Gal4*, we identified 197 downregulated genes and 192 upregulated genes. Sim is one of the downregulated genes. With CUT&RUN experiments, which has higher resolution and is more accurate than ChIP-seq, we also identified multiple Ftz-f1 binding peaks across the genome in follicle cells. In all three replicates, one single peak is identified ~200bp upstream of *sim’s* transcript FBtr0334613, which is the only *sim* transcript expressed in follicle cells and downregulated in *ftz-f1*-knockdown follicles. *de novo* motif analysis identified similar motifs to the canonical Ftz-f1-binding motif derived from previous EMSA studies. These motifs exist in the peak of *sim* with high log-odds score, indicating a direct Ftz-f1-binding site. Therefore, these data suggest that *sim* is a direct target of Ftz-f1. In addition to *sim*, we also identified another 14 genes that are potential direct targets of Ftz-f1. We believe this is a highly conservative estimation due to the stringent criteria we applied. All these data (including one new figure, one supplemental figure, and three supplemental tables) were added to the revised manuscript and significantly strengthened the conclusion of the manuscript.

We understand that it will be stronger to include qRT-PCR, ChIP-PCR, in vitro binding assay, mapping of the entire promoter region of *sim* for follicle cell expression, and in vivo point mutation experiments to further support that sim is a direct target of Ftz-f1 as well as validate some of other direct targets. However, due to the coronavirus pandemic and school lockdown, we were unable to perform any of these experiments. We will pursue these experiments in our future publications.

2) Determining whether sim overexpression is able to rescue phenotypes caused by Ftz-f1 knockdown is a critical test of the idea that sim is genetically downstream of Ftz-f1. The authors attempted this experiment by simultaneously driving Ftz-f1RNAi and sim with Act-Gal4 but the results were complicated because overexpression of sim caused a phenotype earlier in follicle cell differentiation. Thus, the authors should test whether they can circumvent this problem by using another Gal4 driver with a more limited expression pattern, like the Vm26Aa-Gal4 used in Figure 2 or the sim-Gal4 available at BDSC. There are other genetic tricks (Gal80ts etc.) that should allow for the expression of sim in stage 10B follicles when Ftz-f1 in knocked down. These experiments should be explored further and are absolutely critical for the conclusion described in the title.

As reviewers suggested we performed the rescue experiments by simultaneously driving *ftz-f1^RNAi^* and *sim* with *the VM26Aa-Gal4* (Figure 6—figure supplement 2). However, once again we found that overexpression of *sim*, either alone or with *ftz-f1^RNAi2^*, disrupted the transition of follicle cells from stage10A to stage10B, evident by their inability to properly downregulate Hnt or upregulate Cut at stage 10B (Figure 6—figure supplement 2B-C, and F-G). We did see some mild (if any) rescue of Hnt, Cut, and Oamb-RFP expression at stage 14; however, the egg morphology is severely disrupted and dorsal appendage did not form at all when *sim* is overexpressed. Therefore, the appropriate timing and dose of Sim seems critical for follicle cell differentiation and follicle maturation, which prevented us from making a solid conclusion whether ectopic *sim* can rescue *ftz-f1*’s defect from this experiment.

We also examined the expression of *sim3.7-Gal4* (BDSC#26784) in the ovary and found that *sim3.7-Gal4* is not expressed in the ovary at all. This makes perfect sense as the 3.7kb fragment of *sim* in this Gal4 construct does not contain the Ftz-f1 binding site identified in the CUT&RUN experiment and its associated transcript FBtr0082711 is not expressed in follicle cells. As such, there’s currently no available Gal4 that is suitable for this rescue experiment. In recognition of the possibility that *sim* may not be the only target downstream of Ftz-f1 for follicle cell differentiation, we changed our title to “The NR5A nuclear receptor Ftz-f1 promotes follicle maturation and ovulation PARTLY via bHLH/PAS transcription factor Single-minded”.

3) The authors should provide additional data that the induction of ftz-f1 expression is determined by ecdysone levels. For example, can Ftz-f1 be induced by exogenous ecdysone? They should also demonstrate the importance of this temporally restricted expression pattern by expressing Ftz-F1 prematurely and/or delaying its expression.

We also performed experiments to analyze the importance of Ftz-f1’s temporally restricted expression pattern by expressing *ftz-f1* prematurely with flip-out Gal4 clones (Figure 5—figure supplement 2). From these experiments we demonstrated overexpression of *ftz-f1* is able to induce Sim expression prematurely. Furthermore, we found these follicle cells also exhibited defects in transitioning from stage10A to stage10B, indicated by their inability to upregulate Cut and downregulate Hnt expression after stage-10B. Overall, these results suggest that precise temporal expression of Ftz-f1 and its downstream target Sim at stage-10B is critical for proper follicle cell differentiation.